**EMBO** *reports*

# Molecular basis for assembly and activation of the Hook3 – KIF1C complex-dependent transport machinery

Hye Seon Lee [1,6], Daseuli Yu [2,6], Kyoung Eun Baek [3], Ho-Chul Shin [4], Seung Jun Kim [4✉], Won Do Heo [3,5✉] & Bonsu Ku [1✉]

## Abstract

**Microtubule-associated cargo transport, a central process governing the localization and movement of various cellular cargoes, is orchestrated by the coordination of two types of motor proteins (kinesins and dyneins), along with diverse adaptor and accessory proteins. Hook microtubule tethering protein 3 (Hook3) is a cargo adaptor that serves as a scaffold for recruiting kinesin family member 1C (KIF1C) and dynein, thereby regulating bidirectional cargo transport. Herein, we conduct structural and functional analyses of how Hook3 mediates KIF1C-dependent anterograde cargo transport through interaction with KIF1C and PTPN21. We verify the interactions among the three proteins and determine the crystal structure of the Hook3(553–624) – KIF1C(714–809) complex. Subsequent structure-based mutational analysis demonstrates that this complex formation is necessary and sufficient for the interaction between the full-length proteins in HEK293T cells and plays a key role in Hook3- and KIF1C-mediated anterograde transport in RPE1 cells. Thus, this study provides a basis for a comprehensive understanding of how Hook3 cooperates with other components during the initial steps of activation and assembly of the Hook3- and KIF1C-dependent cargo transport machinery.**

**Keywords** Hook3; KIF1C; PTPN21; Transport; Structure
**Subject Categories** Cell Adhesion, Polarity & Cytoskeleton; Structural Biology

## Introduction

Diverse cellular processes, including cell development, polarity, morphology, and migration, are directed by the intracellular movement and distribution of various cellular components, such as mRNA, proteins, vesicles, and organelles. Microtubule-dependent cargo transport serves as a key process that controls the movement and distribution of cellular components; thus, understanding its mechanism is fundamental to elucidate various aspects of cellular processes (Barlan and Gelfand, 2017; Wang et al, 2024). Cargo transport is regulated by two types of molecular motor superfamilies, dyneins and kinesins, which cooperate with each other and with adaptor proteins to form a transport machinery that recruits and delivers cargo along microtubules in both directions (Cui et al, 2019; Hancock, 2014). Numerous studies have sought to elucidate the structural and mechanical details of the transport machinery, which consists of motors, adaptor proteins, and cargoes (Chaaban and Carter, 2022; Lau et al, 2021; Urnavicius et al, 2015). Through structural and biochemical analyses, how the autoinhibited dynein motor is activated by forming a complex with dynactin and a cargo adaptor protein for retrograde (minus-end-directed) transport has been well defined (McKenney et al, 2014; Olenick and Holzbaur, 2019; Reck-Peterson et al, 2018; Schlager et al, 2014; Zhang et al, 2017). In contrast, at least 45 members are known to be contained in the human kinesin superfamily, and how these proteins interact with adaptor or accessory proteins and how the resulting complexes mediate intracellular transport remains incompletely understood (Miki et al, 2001).

Hook microtubule tethering protein 3 (Hook3), a cargo adaptor protein that serves as a crucial scaffold protein linking cargo to microtubule motors, plays a pivotal role in regulating bidirectional early endosome trafficking, managing directional movement, and maintaining Golgi morphology (Bielska et al, 2014; Kendrick et al, 2019; Olenick et al, 2016; Urnavicius et al, 2018). Hook3 comprises four distinct domains: the Hook domain (residues 1–160) that interacts with light intermediate chain 1, a component of the dynein complex (Lee et al, 2018) and three coiled-coil domains (residues 160–239, 240–440, and 480–665, respectively) that are involved in self-oligomerization (Schroeder and Vale, 2016; Siddiqui et al, 2019; Xu et al, 2008). Among these domains, the Hook domain together with the first and second coiled coils serve as the dynein-binding region, whereas the third coiled coil-containing the carboxyl-terminal region interacts with several proteins, including fused toes homolog (FTS), FTS – Hook

[1]Disease Target Structure Research Center, Korea Research Institute of Bioscience and Biotechnology, Daejeon 34141, Korea. [2]Life Science Research Institute, Korea Advanced Institute of Science and Technology, Daejeon 34141, Korea. [3]Department of Biological Sciences, Korea Advanced Institute of Science and Technology, Daejeon 34141, Korea. [4]Critical Diseases Diagnostics Convergence Research Center, Korea Research Institute of Bioscience and Biotechnology, Daejeon 34141, Korea. [5]KAIST Institute for the BioCentury, Korea Advanced Institute of Science and Technology, Daejeon 34141, Korea. [6]These authors contributed equally: Hye Seon Lee, Daseuli Yu. ✉E-mail: ksj@kribb.re.kr; wondo@kaist.ac.kr; bku@kribb.re.kr

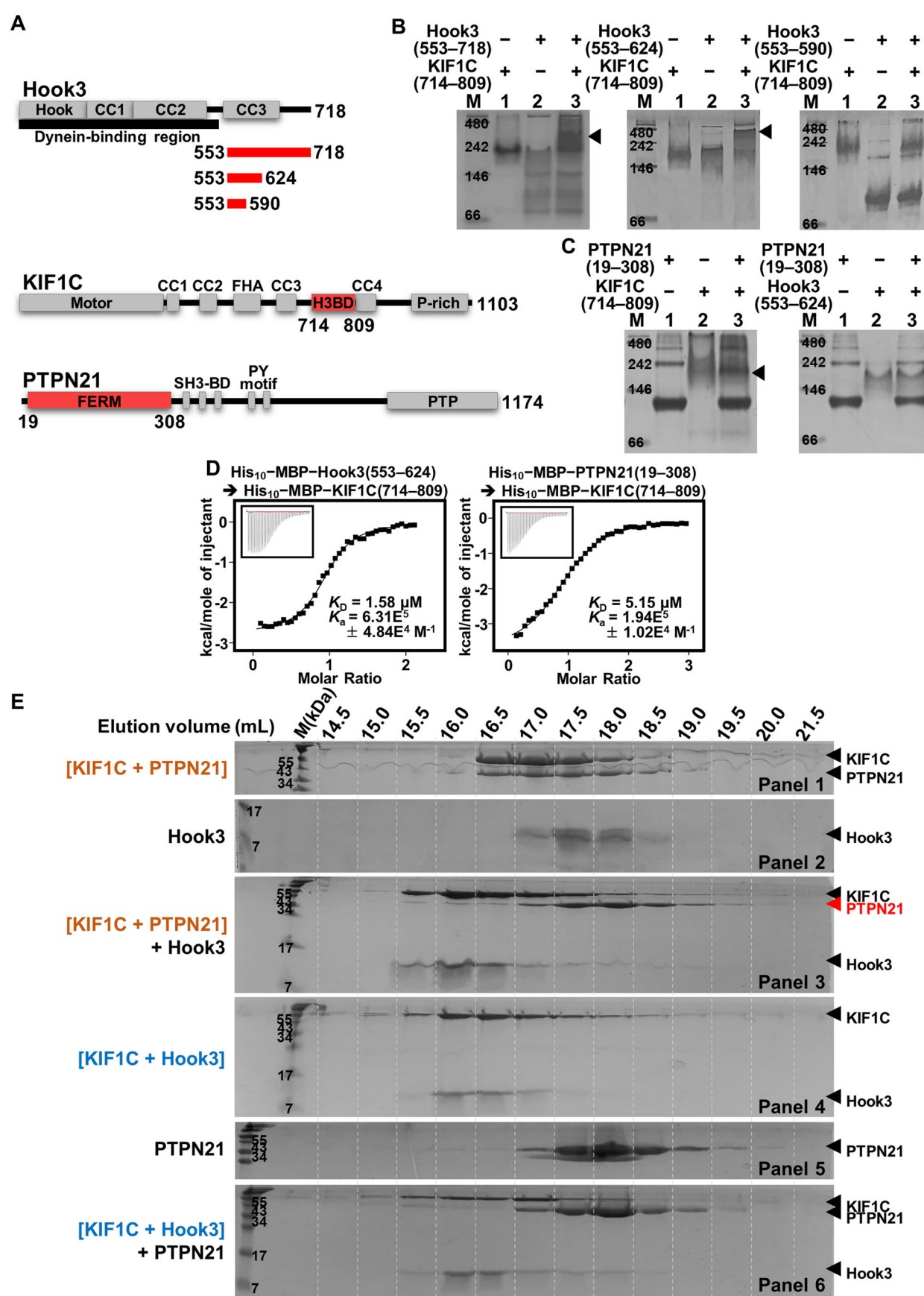

KIF1C, His$_{10}$–MBP–KIF1C(714–809); PTPN21, PTPN21(19–308); Hook3, Hook3(553–624)

◄ **Figure 1. Biochemical verification of interaction among Hook3, KIF1C, and PTPN21.**

(A) Scheme of the PTPN21, KIF1C, and Hook3 domains. The constructs used for the subsequent biochemical analyses were marked in red. (B, C) Native gel electrophoresis. Recombinant proteins were subjected to native-polyacrylamide gel electrophoresis alone or after mixing and 16 h incubation, and then visualized by Coomassie staining. (B) $His_{10} - MBP - KIF1C(714–809)$ and $His_{10} - MBP-Hook3(553–718/553–624/553–590)$; (C) $His_{10} - MBP - PTPN21(19–308)$ and $His_{10} - MBP - KIF1C(714–809)$ or $His_{10} - MBP-Hook3(553–624)$. Arrowheads indicate new bands with high molecular weights appearing in lane 3 of the left and middle panels of (B) and the left panel of (C). M size marker. (D) Binding affinity measurements. ITC was conducted by titrating 0.8 mM $His_{10} - MBP-Hook3(553–624)$ (left) or $His_{10} - MBP - PTPN21(19–308)$ (right) into 0.04 mM $His_{10} - MBP - KIF1C(714–809)$. Curve fittings of the integrated heat per mole of added ligand were used to deduce the $K_D$ values. $K_D$ dissociation constant, $K_a$ association constant. (E) Protein replacement analysis. Recombinant Hook3, KIF1C, and PTPN21 proteins were prepared, mixed as indicated, and then subjected to size-exclusion chromatography in a Superdex™ 200 Increase 10/300 GL column. The fractions eluted from 14.5 to 21.5 mL were analyzed by sodium dodecyl sulfate-polyacrylamide gel electrophoresis and visualized by Coomassie staining. M size marker. Source data are available online for this figure.

−FHIP1B (FHF) complex subunit Hook interacting protein 1B (FHIP1B), and kinesin family member 1 C (KIF1C) (Abid Ali et al, 2025). KIF1C, a kinesin-3 family member with plus-end-directed motility, mediates the anterograde (plus-end-directed) transport of vesicles containing several components, such as α5β1 integrin, 14-3-3, and protein tyrosine phosphatase non-receptor type 21 (PTPN21) (Gabrych et al, 2019). PTPN21 is a cytosolic protein that contains a four-point-one protein, ezrin, radixin, moesin homology (FERM) domain at the amino-terminus and a phosphatase domain at the carboxyl-terminus (Lee et al, 2024). A previous study showed that KIF1C exists as an autoinhibited homodimer, and becomes activated by interacting with the carboxyl-terminal region of Hook3 or the FERM domain of PTPN21 (Siddiqui et al, 2019). Once activated, KIF1C mediates cargo movement from the microtubule-organizing center to the cell periphery, including cell tail tips, which is required for the regulation of a variety of biological processes, such as Golgi organization, podosome dynamics in macrophages, and directional persistence of cell migration (Dorner et al, 1998; Kopp et al, 2006; Theisen et al, 2012). A recent study identified 14 amino acids in the carboxyl-terminal region of KIF1C (residues 794–807) that are involved in binding to Hook3 and recruiting it to the transport machinery (Kendrick et al, 2019). Additionally, another recent study showed that the direct interaction between the stalk region of KIF1C (residues 674–922) and Hook3 (residues 629–728) activates autoinhibited Hook3 (Abid Ali et al, 2025). However, the lack of structural information on these intermolecular interactions has limited our precise understanding of the functionality and biological significance of the Hook3 − KIF1C complex.

In this study, we aimed to investigate the structural and functional basis of Hook3 − KIF1C interaction. We verified the direct intermolecular interaction between Hook3 and KIF1C, and revealed their binding modules, which subsequently led to determination of the crystal structure of Hook3(553–624) bound to KIF1C(714–809) in a 2:2 heterotetrameric form. Using structure-based mutagenesis, we demonstrated that this intermolecular interaction plays a critical role in anterograde transport of the cargo-conjugated protein complex.

## Results

### Hook3 and PTPN21 directly interact with the same region of KIF1C

Based on previous reports that Hook3 and PTPN21 recognize residues 794–807 and 714–809 of KIF1C, respectively (Dorner et al,

1998; Kendrick et al, 2019; Saji et al, 2024), we sought to verify and characterize the interactions among Hook3, KIF1C, and PTPN21 at the molecular level. First, we prepared several recombinant constructs of the three proteins (Fig. 1A), expressed them using the *Escherichia coli* system, and performed native gel electrophoresis to determine whether they directly interact with each other, and to precisely characterize the binding regions. As shown in Fig. 1B (left and middle; marked by arrowheads), a new band appeared when KIF1C(714–809; recently named Hook3-binding domain or H3BD) (Abid Ali et al, 2025), was incubated with Hook3(553–718) or Hook3(553–624) fragments, demonstrating their direct interaction. In contrast, KIF1C(714–809) did not form a complex with Hook3(553–590), implying that the KIF1C-binding region in Hook3 is located at residues 591–624 (Fig. 1B, right). Similarly, a band corresponding to a new complex was detected when KIF1C(714–809) was incubated with the FERM domain of PTPN21 (residues 19–308; referred to as $PTPN21_{FERM}$) (Fig. 1C, left; marked by an arrowhead). Contrastively, no complex formation was observed between PTPN21(19–308) and Hook3(553–624), indicating that these two protein fragments do not interact with each other (Fig. 1C, right). Next, the intermolecular interaction of the H3BD of KIF1C (referred to as $KIF1C_{H3BD}$) with $Hook3_{CC3}$ or $PTPN21_{FERM}$ was verified using isothermal titration calorimetry (ITC). The dissociation constants $(K_D)$ of KIF1C(714–809) with Hook3(553–624) and PTPN21(19–308) were quantified as 1.58 μM and 5.15 μM, respectively (Fig. 1D). Collectively, the biochemical analyses using recombinant proteins confirmed that $KIF1C_{H3BD}$ is the region that binds $Hook3_{CC3}$ and $PTPN21_{FERM}$.

### Hook3 and PTPN21 competitively interact with KIF1C

Our biochemical data raised the question of whether Hook3 and PTPN21 form a ternary complex with KIF1C or compete to bind KIF1C. To address this issue, we performed a protein replacement analysis by subjecting the three recombinant protein constructs to size-exclusion chromatography (Fig. 1E). First, when $His_{10} - MBP - KIF1C(714–809)$ and PTPN21(19–308) were subjected to the Superdex™ 200 Increase 10/300 GL column, the two proteins eluted together at an elution volume of 17.0 mL (Fig. 1E, panel 1). Next, we analyzed recombinant Hook3(553–624) protein, which eluted at an elution volume of 17.5 mL in the same column (Fig. 1E, panel 2). Hook3 was then mixed with the KIF1C − PTPN21 complex sample at the 1:1:1 molar ratio, and the resulting mixture was analyzed by size-exclusion chromatography after overnight incubation. As shown in Fig. 1E (panel 3), Hook3 co-eluted with KIF1C at an elution volume of 16.0 mL, whereas PTPN21 was distinct from the original complex and eluted at an elution volume of

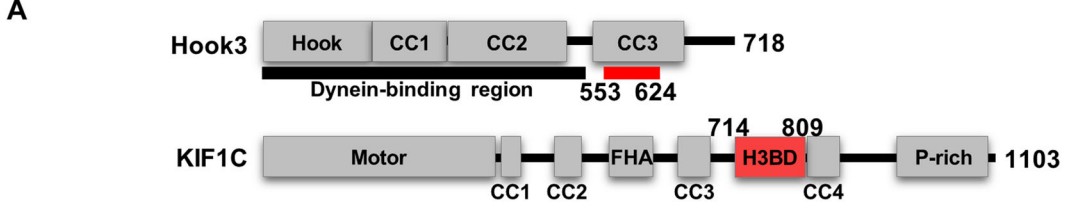

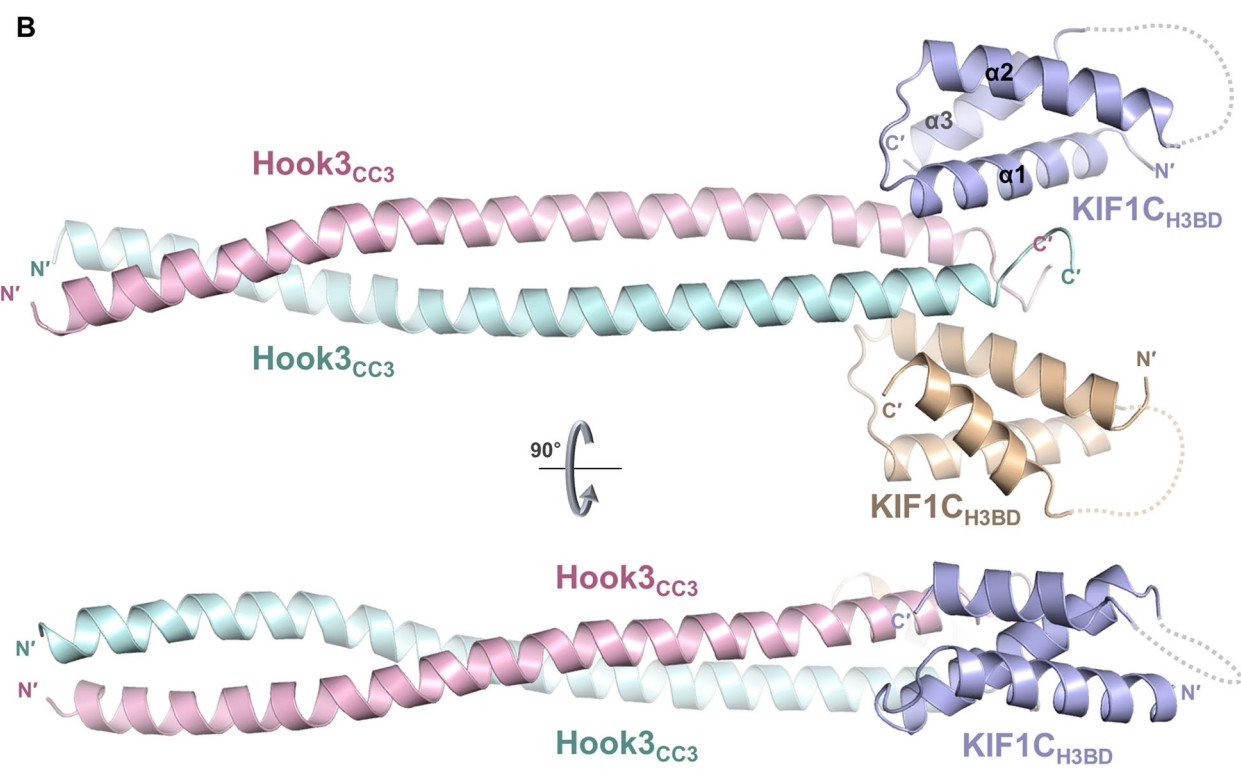

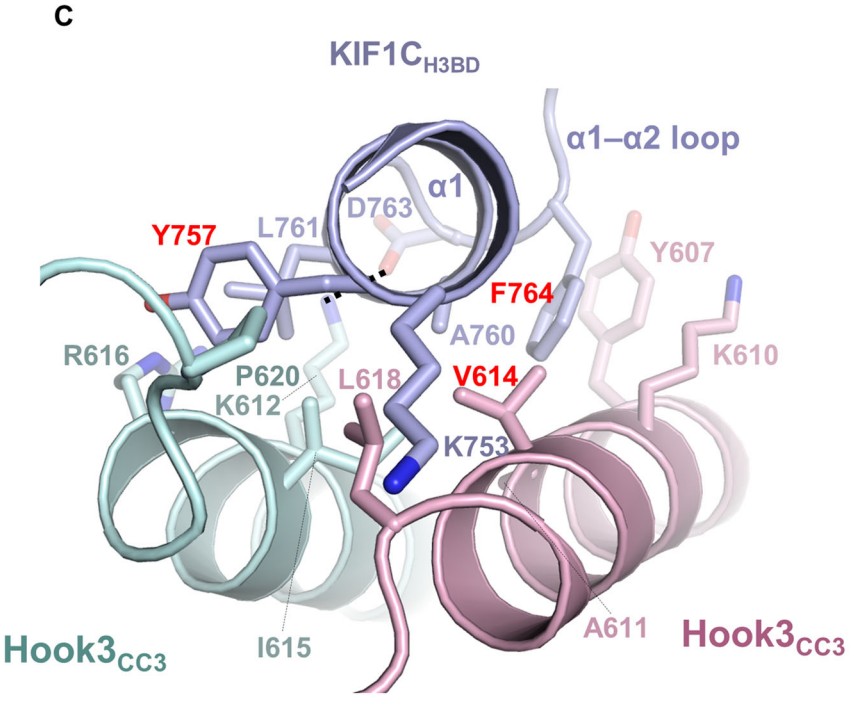

◀ **Figure 2.  Structural analysis of the Hook3 – KIF1C complex.**

(**A**) Schematic diagram showing the domains of Hook3 and KIF1C. The two truncated constructs used for the crystallization were marked in red. (**B**) Overall structure of Hook3(553–624; pink and mint) bound to KIF1C(714–809; navy and beige) shown as ribbon representation. N′ and C′ indicate the amino- and carboxyl-terminus of each polypeptide, respectively. Dashed lines linking α2 and α3 of KIF1C represent invisible regions in the crystal structure owing to poor electron density. (**C**) Detailed view of the intermolecular interaction between KIF1C (navy) and two Hook3 proteins (pink and mint). Residues involved in the complex formation are represented as sticks with labels. The three amino acids selected for preparation of binding-disrupting mutant proteins (Tyr757 and Phe764 of KIF1C; Val614 of Hook3) are shown in red.

18.0 mL (marked by a red arrowhead). These results suggest that Hook3$_{CC3}$ replaces KIF1C$_{H3BD}$-bound PTPN21$_{FERM}$ to constitute a novel 1:1 complex with KIF1C$_{H3BD}$. Subsequently, we prepared two additional samples: PTPN21(19–308) and the complex formed by His$_{10}$ – MBP – KIF1C(714–809) and Hook3(553–624). When they were subjected to the Superdex™ 200 Increase 10/300 GL column separately (Fig. 1E, panels 4 and 5) or after mixing and incubating overnight (Fig. 1E, panel 6), we found that the elution volumes of PTPN21 (18.0 mL) and the Hook3 – KIF1C complex (16.0 mL) did not change, indicating that PTPN21$_{FERM}$ was hardly able to replace KIF1C$_{H3BD}$-bound Hook3$_{CC3}$ (Fig. 1E, panel 6). Taken together, these results suggest that Hook3$_{CC3}$ and PTPN21$_{FERM}$ interact with KIF1C$_{H3BD}$ in a competitive manner, rather than by forming a ternary complex. Notably, Hook3 replaced PTPN21-bound KIF1C to constitute a new complex with KIF1C (Fig. 1E, panel 3), consistent with our binding affinity measurements showing that KIF1C$_{H3BD}$ binds to Hook3$_{CC3}$ ($K_D$ of 1.58 μM) with a considerably higher affinity than to PTPN21$_{FERM}$ ($K_D$ of 5.15 μM) (Fig. 1D).

## Crystal structure of Hook3 bound to KIF1C in a 2:2 heterotetrameric complex

Next, we attempted to crystallize Hook3(553–624) and KIF1C(714–809), separately and as a complex (Fig. 2A; marked in red). The crystal structures of Hook3(553–624) in an apo form (Fig. EV1A) and in a complex with KIF1C(714–809) (Fig. 2B) were determined to resolutions of 2.7 Å and 3.0 Å, respectively (Table EV1). Even though Hook3 exists in a parallel coiled-coil form in both structures (Figs. 2B and EV1A), the two Hook3 structures did not exactly match up when superimposed (Fig. EV1B). It was presumably due to crystal packing interactions among the carboxyl-terminal regions of the Hook3(553–624) protomers in the Hook3 single structure, which artificially twisted the direction of its coiled coils (Fig. EV1C). Therefore, the entire atomic-level molecular analyses described henceforth were conducted using the complex structure, in which Hook3(553–624) and KIF1C(714–809) constituted a 2:2 heterotetramer (Fig. 2B).

In this complex, residues 553–618 of Hook3, which constitute the parallel coiled coil, consist of two distinct types of heptad repeats (Fig. EV2). In this study, we defined them as types I (residues 553–601, seven repeats) and II (residues 602–615, two repeats) based on the position of the coiled-coil interface-forming hydrophobic residues: type I at the third and sixth, and type II at the third and seventh (Fig. EV2A). As shown in Fig. EV2B, the coiled-coil conformation of Hook3 is maintained by an elaborate network of hydrophobic interactions. This homodimeric Hook3 coiled-coil acts as a scaffold for complex formation with two KIF1C(714–809) molecules, in which residues 742–786 and 793–807 are visible in our complex crystal structure (Fig. 2B). KIF1C(714–809) forms an antiparallel three-helical bundle

consisting of α1 (residues 744–762), α2 (residues 768–785), and α3 (residues 795–807) (Fig. 2B). The protein folding of KIF1C(714–809) is sustained by intensive intramolecular hydrophobic interactions between the residues constituting the three α-helices, which include Lys748, Val752, Ile755, and Val759 from α1; Phe764 from the α1–α2 loop; Ile771, Ala775, Ala776, and Met779 from α2; Trp796, Val799, Ala800, Val803, Trp804, and Val807 from α3 (Fig. EV3A).

## Structural analysis of the intermolecular association between Hook3 and KIF1C

Our crystal structure revealed several key hydrophobic residues from the α1 helix and the α1–α2 loop of KIF1C that directly contribute to the complex formation, in which each single KIF1C protomer recognizes both molecules of homodimeric Hook3 (Fig. 2C). Specifically, the main Hook3-binding interface of KIF1C is constituted by Lys753, Tyr757, Ala760, and Leu761 from α1; Asp763 and Phe764 from the α1–α2 loop, which are associated with Tyr607, Lys610, Ala611, Val614, and Leu618 of one Hook3 molecule and Lys612, Ile615, Arg616, and Pro620 of another Hook3 protomer that coordinately form a large extended hydrophobic surface on the Hook3 homodimer (Figs. 2C and EV3B). The complex formation is also reinforced by electrostatic interaction between Lys612 of Hook3 and Asp763 of KIF1C (Fig. 2C).

To verify the relevance of our structure, complex formation-disrupted mutant versions of Hook3 and KIF1C were additionally constructed: Hook3(553–624; V614E) and KIF1C(714–809; Y757A · F764A). These mutant proteins were too unstable compared to their wild-type counterparts to be purified in our experimental conditions, and thus deemed unsuitable for ITC that is generally performed at 25 °C. Therefore, we performed native gel electrophoresis at 4 °C using the wild-type and mutant recombinant Hook3 and KIF1C proteins (Fig. 3A, lanes 1–4) and their mixtures incubated for 16 h at a molar ratio of 1:1 (Fig. 3A, lanes 5–8). A new band with a higher molecular weight appeared in lane 5 containing a mixture of wild-type Hook3(553–624) and KIF1C(714–809) (Fig. 3A; marked by an arrowhead). In contrast, complex formation was not detected in lanes 6–8, which contained at least one mutant protein (Fig. 3A). Furthermore, we found that alanine substitutions of Tyr757 and Phe764, the two key residues of KIF1C in binding Hook3, also led to defects in the intermolecular interaction with PTPN21 (Fig. 3B). These results suggested that Tyr757 and Phe764 of KIF1C are critically involved in its binding to both Hook3$_{CC3}$ and PTPN21$_{FERM}$, which competitively interact with KIF1C$_{H3BD}$ (Fig. 1E).

To further validate the physical interaction between Hook3 and KIF1C, we performed a co-immunoprecipitation assay in HEK293T cells co-expressing Myc-tagged full-length Hook3(wild-type/ V614E) and Flag-tagged full-length KIF1C(wild-type/Y757A ·

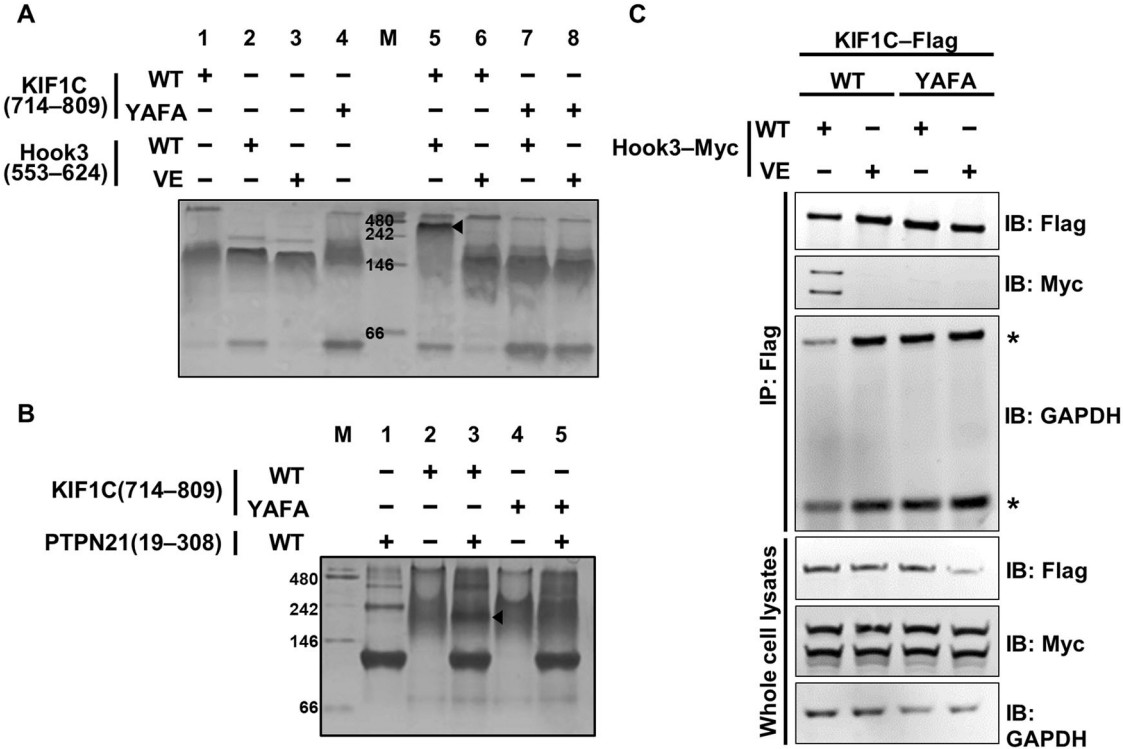

**Figure 3. Biochemical analysis of intermolecular binding among Hook3, KIF1C, and PTPN21.**

(A, B) Native gel electrophoresis. (A) Recombinant $His_{10} - MBP - KIF1C$(714–809; WT or Y757A $\cdot$ F764A) and $His_{10} - MBP - Hook3$(553–624; WT or V614E) were subjected to native gel electrophoresis individually (lanes 1–4) or after mixing and 16 h incubation (lanes 5–8) and then visualized by Coomassie staining. (B) Recombinant $His_{10} - MBP - KIF1C$(714–809; WT or Y757A $\cdot$ F764A) and $His_{10} - MBP - PTPN21$(19–308; WT) were subjected to native gel electrophoresis individually (lanes 1, 2, and 4) or after mixing and 16 h incubation (lanes 3 and 5) and then visualized by Coomassie staining. The arrowhead indicates the new band with a higher molecular weight in lane 3. M size marker, WT wild-type, YAFA Y757A $\cdot$ F764A, VE V614E. (C) Co-immunoprecipitation analysis. Intermolecular binding between full-length KIF1C(WT or Y757A $\cdot$ F764A)—Flag—EGFP and Hook3(WT or V614E)—Myc—FuRed transiently expressed in HEK293T cells was analyzed by immunoprecipitation and immunoblotting as indicated. Glyceraldehyde 3-phosphate dehydrogenase (GAPDH) was used as a protein loading control. Asterisks indicate heavy or light immunoglobulin chains. WT wild-type, YAFA Y757A $\cdot$ F764A, VE V614E, IP immunoprecipitation, IB immunoblotting. Source data are available online for this figure.

F764A). As shown in Fig. 3C, the intermolecular interaction between full-length Hook3 and KIF1C was distinctively detected between the wild-type proteins (Fig. 3C, lane 1) but was impaired when mutations that abrogate $Hook3_{CC3} - KIF1C_{H3BD}$ binding were introduced into either of the two proteins (Fig. 3C, lanes 2–4). Therefore, these results demonstrated the significance of the intermolecular association shown in the $Hook3_{CC3} - KIF1C_{H3BD}$ crystal structure and its critical contribution to the complex formation between their full-length forms.

## Tail enrichment of Hook3 and KIF1C is enhanced following complex formation

Hook3 is a cargo adaptor that acts as a scaffold protein to recruit the two motor proteins, dynein and KIF1C, which are involved in microtubule-mediated cargo transport (Kendrick et al, 2019). Among them, KIF1C is responsible for the intracellular transport of specific cargo from the cytoplasm to peripheral regions (Kopp et al, 2006; Siddiqui et al, 2019). KIF1C is enriched in the tail tip regions of retinal pigment epithelial-1 (RPE1) cells, and therefore they were used for tracking KIF1C-mediated subcellular transport in previous studies (Siddiqui et al, 2022; Siddiqui et al, 2019;

Theisen et al, 2012) and this study. To assess the importance of the interaction between Hook3 and KIF1C at a cellular level, we investigated the effects of binding-disrupting mutations (Y757A $\cdot$ F764A in KIF1C and V614E in Hook3; see Figs. 2 and 3) on the subcellular distribution of Hook3 and KIF1C. When expressed individually for 16 h in RPE1 cells, we found that wild-type and mutant KIF1C were enriched in the tail tips at tail/cytoplasm ratios of $2.7 \pm 0.3$ and $3.4 \pm 0.3$, respectively (Fig. EV4). In contrast, wild-type and mutant Hook3 were not, as their tail/cytoplasm ratios were $1.1 \pm 0.1$ for both constructs (Fig. EV4). These results imply the following: KIF1C tends to accumulate in the tail region to a higher degree than Hook3, and the presence or absence of the binding-disrupting mutations does not significantly affect the distribution of Hook3 and KIF1C when they are expressed separately.

However, the effects of the binding-disrupting mutations were obvious when the two proteins were expressed together in RPE1 cells (Fig. 4A). When wild-type Hook3 and KIF1C were co-expressed, the tail/cytoplasm ratio of KIF1C was $3.6 \pm 0.4$ (Fig. 4B). In contrast, the corresponding ratios were $2.0 \pm 0.2$, $2.4 \pm 0.3$, and $2.1 \pm 0.2$ in cells expressing a combination of the two proteins with at least one binding-defective mutant form (Fig. 4B).

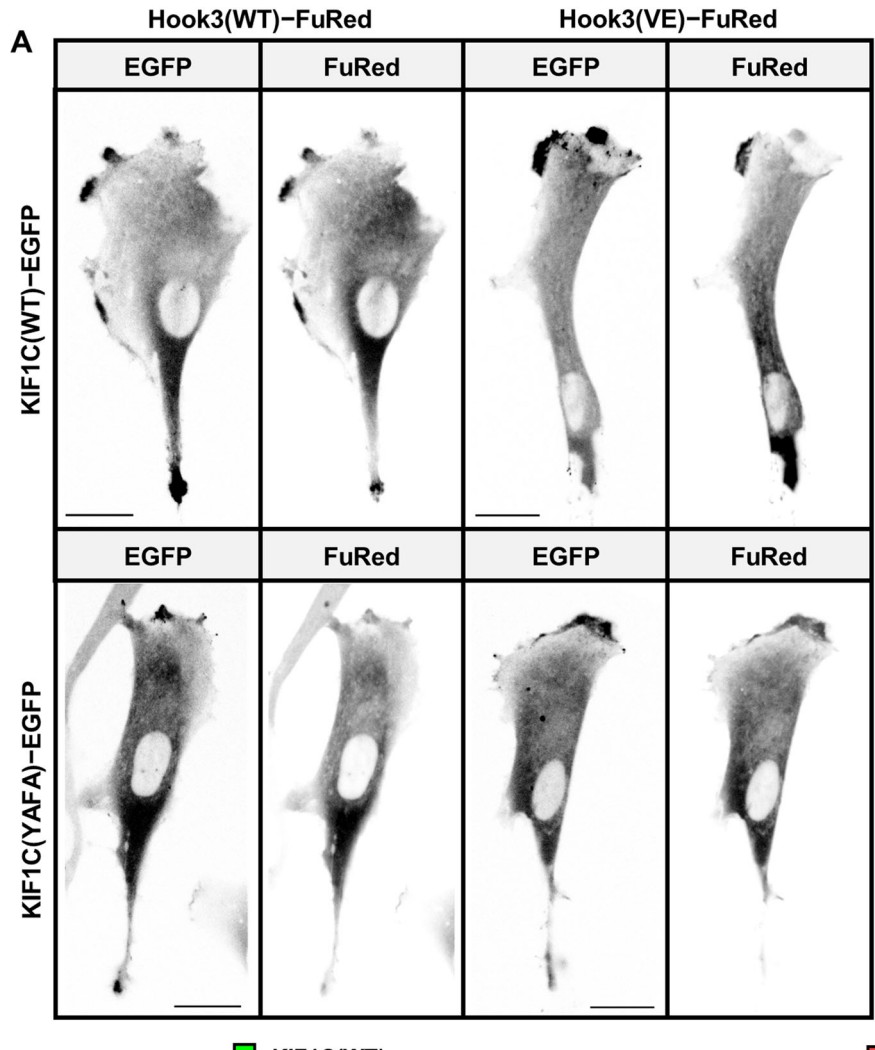

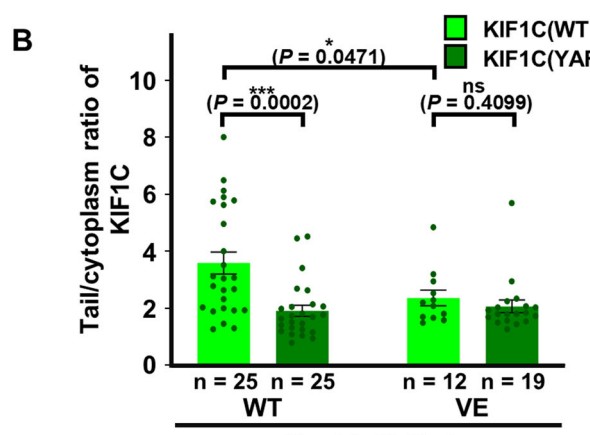

| Hook3 | KIF1C | tail/cytoplasm ratio of KIF1C |
|---|---|---|
| wild-type | wild-type | 3.6 ± 0.4 |
|  | Y757A·F764A | 2.0 ± 0.2 |
| V614E | wild-type | 2.4 ± 0.3 |
|  | Y757A·F764A | 2.1 ± 0.2 |

| Hook3 | KIF1C | tail/cytoplasm ratio of Hook3 |
|---|---|---|
| wild-type | wild-type | 1.3 ± 0.1 |
|  | Y757A·F764A | 0.7 ± 0.1 |
| V614E | wild-type | 0.7 ± 0.1 |
|  | Y757A·F764A | 0.7 ± 0.1 |

◀ 

◀ **Figure 4.   Tail enrichment of Hook3 and KIF1C in RPE1 cells.**

(A) Representative images of RPE1 cells co-expressing wild-type or mutant KIF1C—Flag—EGFP and Hook3—Myc—FuRed at 16 h post transfection. Scale bars, 10 μm. (B, C) Tail enrichment of KIF1C—Flag—EGFP (B) and Hook3—Myc—FuRed (C) upon co-expression in RPE1 cells. Tail/cytoplasm ratio of each protein is presented as histogram (top) and in tabular forms (bottom). Values are means ± standard error of the mean. ns not significant; *P < 0.05; ***P < 0.005 by the Student's two-tailed t test. WT wild-type, YAFA Y757A · F764A, VE V614E. Source data are available online for this figure.

Likewise, the tail/cytoplasm ratio of Hook3 was $1.3 \pm 0.1$ when wild-type forms of the two proteins were co-expressed, whereas the ratio in cells expressing at least one mutant form was half that ($0.7 \pm 0.1$ in all the three cells; Fig. 4C). Thus, these results suggest that Hook3 and KIF1C form a complex upon co-expression, which facilitates their transport and tail enrichment in an intermolecular binding-dependent manner.

## Disruption of Hook3 − KIF1C interaction significantly impairs Hook3-mediated cargo transport

Next, we sought to determine the effects and significance of the intermolecular interaction between Hook3 and KIF1C on intracellular cargo transport. Hence, we used the chemically inducible FK506-binding protein (FKBP) − FKBP12-rapamycin binding domain (FRB) heterodimerization system, which has been widely used for tracking kinesin motor-directed intracellular movement of cargoes, such as mitochondria, endosomes, and peroxisomes (Budaitis et al, 2021; Lipka et al, 2016; Patel et al, 2021; Serra-Marques et al, 2020; Siddiqui et al, 2022). We co-expressed the following three constructs in RPE1 cells: Hook3(wild-type/V614E)−FuRed−FRB, KIF1C(wild-type/Y757A · F764A)−mTagBFP2, and FKBP − EGFP−monoamine oxidase A (MoA) for tagging intracellular mitochondria (Fig. 5A). Treatment with rapamycin induced the formation of the FKBP−rapamycin−FRB complex, which led to the recruitment of FKBP − EGFP−MoA-tagged mitochondria to the Hook3 − KIF1C complex as a cargo for intracellular anterograde delivery (Fig. 5A). Using this system, we traced the movement of Hook3 − KIF1C complex-conjugated mitochondria (Fig. 5B; Movies EV1−EV4). After 10 min of rapamycin treatment of RPE1 cells co-expressing wild-type Hook3 and KIF1C, we found that EGFP-tagged mitochondria were enriched in the tail region (Fig. 5B, marked by colored rectangles; Movie EV1). As expected, intensity analysis of the tail region verified that the distribution of EGFP-tagged mitochondria overlapped well with that of the two wild-type proteins in these cells (Fig. 5C). In contrast, when the binding-disrupting mutations were introduced into at least one of the two proteins, EGFP-tagged mitochondria appeared to be enriched in the perinuclear region, but not in the tail tips, and its distribution overlapped with that of Hook3 but not with that of KIF1C (Fig. 5B; Movies EV2−EV4). Consistently, significant enrichment of EGFP-tagged mitochondria in the tail region of RPE1 cells was only detected when the two wild-type proteins were co-expressed (Fig. 5D). Next, the distribution of EGFP-tagged mitochondria was quantified by profiling line intensities from the center of the cell to the tips at two min intervals. From -2 min to 16 min after rapamycin treatment, EGFP-tagged mitochondria became enriched in the tail tips in a time-dependent manner only when the wild-type forms of the two proteins were co-expressed, which indicates that anterograde delivery requires the formation of the Hook3 − KIF1C complex

(Fig. 5E). Collectively, these results indicate that the interaction between Hook3$_{CC3}$ and KIF1C$_{H3BD}$, which was determined by our structural analysis, plays a critical role in the cytoplasm-to-edge cargo transport mediated by the Hook3 − KIF1C complex.

# Discussion

## Structural determination and analysis of the Hook3 − KIF1C complex

Proper association between motor and cargo adaptor proteins is a critical step in the assembly of the transport machinery. Therefore, it is crucial to precisely identify the binding regions and intermolecular interactions between these proteins at the atomic level. Hook3, a cargo adaptor protein, is involved in both anterograde and retrograde vesicle transport, because it interacts not only with dynein but also with KIF1C (Kendrick et al, 2019). To date, several structural studies have provided a molecular basis for understanding the mechanism of Hook3-associated intracellular trafficking. For instance, the structure of the Hook domain of Hook3 bound to a fragment derived from dynein light intermediate chain 1 was determined by X-ray crystallography (Lee et al, 2018), and the complex architecture of the dynactin−dynein machinery containing the Hook3 coiled-coil fragment was elucidated using cryo-electron microscopy (cryo-EM) (Lau et al, 2021; Urnavicius et al, 2018). However, to the best of our knowledge, no structural information regarding the intermolecular interaction between Hook3 and KIF1C at an atomic level has been reported. In this study, we performed structural, biochemical, and cellular analyses of the Hook3 − KIF1C complex, and investigated its biological function. A combination of biochemical and structural approaches revealed that Hook3$_{CC3}$ contains a binding region which interacts with the helical bundle structure of KIF1C$_{H3BD}$, thereby forming a 2:2 heterotetrameric complex in which two monomeric units of KIF1C are assembled with the Hook3 homodimer (Figs. 1, 2, EV1, EV2, and EV3). In this complex, the binding between the two proteins was found to mainly rely on hydrophobic interactions (Fig. 2C), which were used for the construction of binding-disrupting single (V614E in Hook3) and double (Y757A and F764A in KIF1C) mutants for further verification. Introduction of these mutations not only impaired the interaction between the recombinant Hook3(553–624) and KIF1C(714–809) fragments (Fig. 3A), but also abrogated the binding between the full-length constructs expressed in HEK293T cells (Fig. 3C). Moreover, tail enrichment of Hook3 and KIF1C (Fig. 4) and transport of the Hook3-conjugated mitochondria cargo to the cell tips (Fig. 5) occurred only when the two proteins were co-expressed as wild-type, but not when either one of them was expressed in the binding-disrupted mutant form. These data indicate that the intermolecular

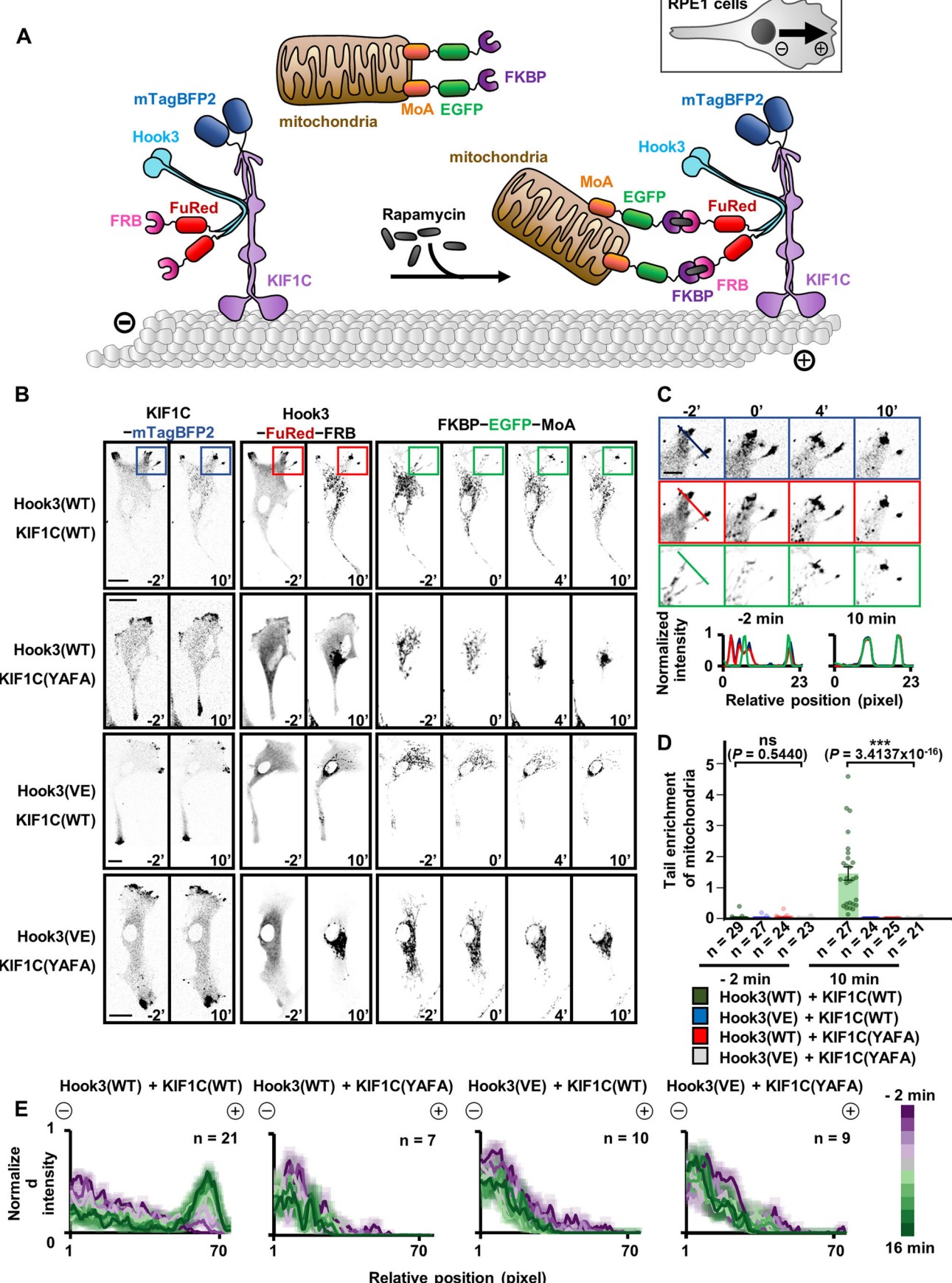

**Figure 5. Interaction with KIF1C facilitates Hook3-mediated anterograde cargo transport in RPE1 cells.**

(A) Schematic representation of the chemically inducible cargo delivery analysis. Rapamycin treatment leads to recruitment of FKBP — EGFP—MoA-tagged mitochondria to FuRed—FRB-tagged Hook3 on microtubules. (B) Live-cell images of RPE1 cells co-expressing FKBP — EGFP—MoA, KIF1C—mTagBFP2, and Hook3—FuRed—FRB in the wild-type or mutant forms. Numbers with an apostrophe represent the time lapses (in min) after 500 nM rapamycin treatment. Colored rectangles represent the cell tips analyzed in (C). Scale bars, 10 μm. WT wild-type, YAFA Y757A · F764A, VE V614E. (C) Magnified images of boxes marked in (B) with intensity profiles along the indicated lines. Each intensity on one-line profile was normalized to that of the maximum value of the profile. (D) Tail enrichment of FKBP — EGFP—MoA-tagged mitochondria before (−2 min) and after (10 min) 500 nM rapamycin treatment. Tail/cytoplasm ratio of tagged mitochondria under each co-expression condition was analyzed, and is presented as dot plots. Values are means ± standard error of the mean. ns, not significant; ***$P < 0.0001$ by one-way ANOVA. WT wild-type, YAFA Y757A · F764A, VE V614E. (E) Time-course intensity profiles of the tail/cytoplasm ratio of FKBP — EGFP—MoA-tagged mitochondria after 500 nM rapamycin treatment. Each intensity on one-line profile was normalized to that of the maximum value of the profile. WT wild-type, YAFA Y757A · F764A, VE V614E. Source data are available online for this figure.

interaction determined by the crystal structure (Fig. 2) is necessary and sufficient for Hook3 − KIF1C complex formation.

## Crystal structure revealed precise atomic information of the complex assembly

Owing to the lack of the atomic structure of the Hook3 − KIF1C complex, several previous studies have attempted to determine their intermolecular interaction using diverse biochemical approaches (Abid Ali et al, 2025; Kendrick et al, 2019). For example, it was assumed that residues 794–807 of KIF1C, which correspond to the α3 helix of our KIF1C$_{H3BD}$ structure (Fig. EV3A), constitute the Hook3-binding region based on the co-immunoprecipitation analysis and single-molecule motility assay using the KIF1C construct in which those residues were truncated (Kendrick et al, 2019). However, according to the complex structure determined in this study, the α1 helix and α1–α2 loop, but not α3 of KIF1C, are significantly involved in direct association with Hook3 (Fig. 2C). Instead, the α3 helix seems indispensable for maintaining the H3BD conformation by mediating tight intramolecular hydrophobic interactions in which Trp796, Val799, Ala800, Val803, Trp804, and Val807 of α3 are involved (Fig. EV3A). Therefore, truncation of residues 794–807 in KIF1C results in severe destabilization of the helical bundle structure of KIF1C$_{H3BD}$, which accounts for its impaired interaction with Hook3 reported in the previous study (Kendrick et al, 2019). In the recent study, Abid Ali et al presented the cryo-EM structure of the FHF complex, in which they modeled the Hook3 − KIF1C complex using Alpha-Fold2, followed by cross-linking mass spectrometric analysis to specify intermolecular interactions between the two proteins (Abid Ali et al, 2025). Although their result was consistent with our crystal structure in which KIF1C$_{H3BD}$ recognized the tail region of Hook3$_{CC3}$, we found two critical discrepancies between our structure and their model. First, they described that one KIF1C$_{H3BD}$ protomer interacts with a single Hook3 molecule in the previous AlphaFold2-based model (Abid Ali et al, 2025), whereas our crystal structure demonstrated that both Hook3 protomers participated in the interaction with a single KIF1C$_{H3BD}$ molecule (Fig. 2C). Second, in our crystal structure, the KIF1C$_{H3BD}$-binding interface of Hook3 was restricted to residues 607–622, which was supported by co-immunoprecipitation analysis showing that a single V614E substitution was sufficient to abrogate complex formation between the full-length Hook3 and KIF1C constructs (Fig. 3C). Thus, among the eight Hook3 residues that were assumed to be in proximity to KIF1C$_{H3BD}$ based on cross-linking mass spectrometric analysis (Abid Ali et al, 2025), only Lys606, Lys610, Lys612, and Lys621 of Hook3 appear adjacent to KIF1C$_{H3BD}$ (Fig. EV3C, Abid

Ali et al, 2025; Data ref: Abid Ali et al, 2025). These data collectively demonstrate the significance of atomic level-experimental structure determination using X-ray crystallography, which provides a basis for understanding the constitution and working mechanism of protein machinery with high accuracy.

## Structural and biochemical analysis of association among Hook3, KIF1C, and PTPN21

Although the results of a previous study (Abid Ali et al, 2025) and our structural analysis (Fig. 2) clearly demonstrate that KIF1C$_{H3BD}$ is the region that binds Hook3$_{CC3}$, PTPN21$_{FERM}$ was also reported to interact with the same region of KIF1C, as proven by co-immunoprecipitation and western blot analysis (Dorner et al, 1998; Siddiqui et al, 2019). Moreover, both interactions mediated by KIF1C$_{H3BD}$ led to the release and activation of autoinhibited KIF1C (Dorner et al, 1998; Siddiqui et al, 2019), suggesting that Hook3 and PTPN21 functionally cooperate to facilitate KIF1C-mediated cargo transport. In this study, we verified and quantified the binding interaction of KIF1C$_{H3BD}$ with Hook3$_{CC3}$ ($K_D$ of 1.58 μM) and PTPN21$_{FERM}$ ($K_D$ of 5.15 μM) using native gel electrophoresis (Fig. 1B,C) and ITC (Fig. 1D). Further size-exclusion chromatography analysis demonstrated that PTPN21$_{FERM}$ and Hook3$_{CC3}$ compete to bind KIF1C$_{H3BD}$ rather than they form a ternary complex (Fig. 1E), and KIF1C$_{H3BD}$ preferentially interacts with Hook3$_{CC3}$ over PTPN21$_{FERM}$ (Fig. 1E), consistent with their distinctive binding affinities measured by ITC (Fig. 1D). Alanine substitution of KIF1C Tyr757 and Phe764, the two key residues for the Hook3$_{CC3}$ − KIF1C$_{H3BD}$ complex formation (Fig. 2), abrogated intermolecular interaction of KIF1C$_{H3BD}$ not only with Hook3$_{CC3}$ but also with PTPN21$_{FERM}$ (Fig. 3A,B). These results imply that the hydrophobic surface of KIF1C serves as a platform for its competitive binding to Hook3 and PTPN21, which is presumably required for KIF1C activation.

Previously, it was reported that the interaction between PTPN21 and KIF1C was enhanced by the c-Src kinase-mediated phosphorylation of four tyrosine residues in KIF1C: Tyr 654, Tyr671, Tyr726, and Tyr757 (Saji et al, 2022). Interestingly, c-Src is known as a PTPN21 substrate and is activated through dephosphorylation by PTPN21 (Cardone et al, 2004; Carlucci et al, 2008). Furthermore, among the four KIF1C tyrosine residues, Tyr757 was revealed as the key residue for the Hook3$_{CC3}$ − KIF1C$_{H3BD}$ complex formation via our crystal structure determination, and thus its phosphorylation by c-Src is expected to impair the intermolecular binding (Fig. 2C). Taken together, the FERM domain-dependent association of PTPN21 with KIF1C appears to be supported by its amino-terminal phosphatase domain, which can dephosphorylate and activate c-Src. Once

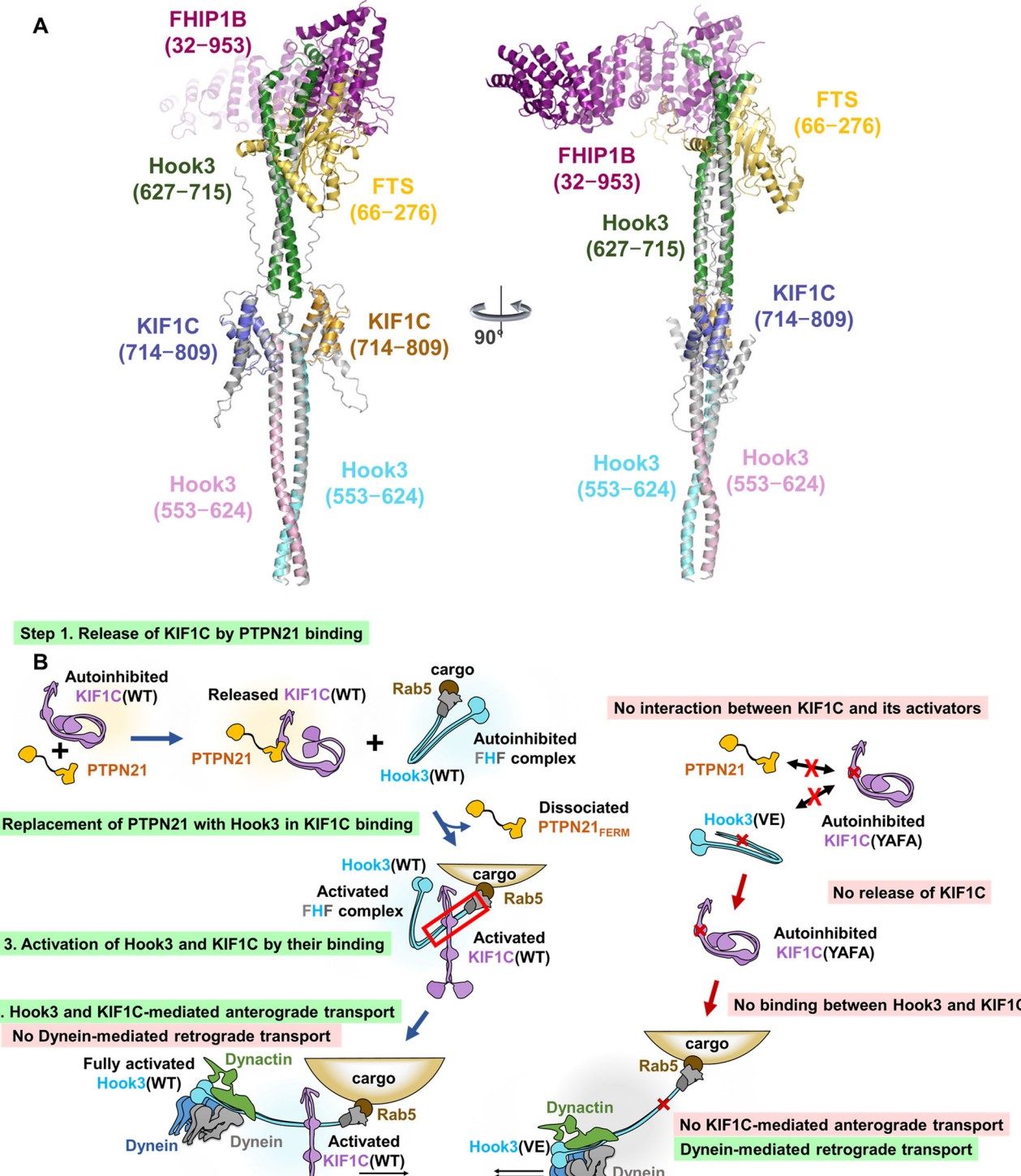

**Figure 6. Models of the Hook3- and KIF1C-dependent assembly and activation of the cargo transport machinery.**

(A) Structural model of the KIF1C-bound FHF complex constructed by superimposition of the three complexes shown in Fig. EV5: FTS(66–276) −Hook3(627–715) − FHIP1B(32–953) complex (Fig. EV5A; PDB code: 8QAT) (Abid Ali et al, 2025; Data ref: Abid Ali et al, 2025), Hook3(553–624) − KIF1C(714–809) complex (Fig. EV5B; PDB code: 9KO8), and AlphaFold2-predicted Hook3(553–718) − KIF1C(714–840) complex. (B) Molecular model depicting the assembly and activation of the Hook3- and KIF1C-dependent anterograde cargo transport machinery (Abid Ali et al, 2025; Siddiqui et al, 2019). WT wild-type, YAFA Y757A · F764A, VE V614E.

activated, c-Src phosphorylates KIF1C, promoting the PTPN21−KIF1C interaction while presumably inhibiting the Hook3 − KIF1C binding. Therefore, we propose that the two domains of PTPN21 functionally cooperate to precisely regulate its interaction with KIF1C, which requires further investigation for verification.

## Structural and mechanistic model of the Hook3- and KIF1C-dependent cargo transport machinery

According to our crystal structure analysis, the Hook3 − KIF1C complex contains the third coiled coil (residues 553–624) of Hook3, whereas the cryo-EM structure of the FHF complex reported by Abid Ali et al (Abid Ali et al, 2025) includes its carboxyl-terminal domain (residues 627–715) (Figs. EV5A and EV5B, Abid Ali et al, 2025; Data ref: Abid Ali et al, 2025). To present the two complexes in a combined form, we performed AlphaFold2-based modeling of the Hook3(553–718) − KIF1C(714–840) complex (Fig. EV5C), and then used the resulting structural model as a frame on which the two experimentally determined structures were superimposed. The final model, shown in Fig. 6A (Abid Ali et al, 2025; Data ref: Abid Ali et al, 2025), demonstrates a precise molecular view of how Hook3 orchestrates the arrangement of these protein units and the assembly of the complex, providing a fascinating snapshot of the initial step of KIF1C binding-induced activation of the autoinhibited FHF complex. Based on this quadruple complex model obtained from the two studies, we propose a model describing the detailed procedure of assembly and activation of the Hook3- and KIF1C-dependent cargo machinery by combining data provided by our and previous studies (Abid Ali et al, 2025; Siddiqui et al, 2019). The results of diverse biochemical investigations, including cross-linking mass spectrometric analysis and cryo-EM structure determination (Abid Ali et al, 2025; Data ref: Abid Ali et al, 2025) have suggested that KIF1C and the FHF complex, which serves as a Rab5-recruiting cargo adaptor, exist in autoinhibited forms prior to activation (Fig. 6B, left; Step 1). PTPN21 interacts with KIF1C via PTPN21$_{FERM}$ and KIF1C$_{H3BD}$ binding, which promotes the release of KIF1C from its motor domain and stalk region-bound autoinhibited state (Fig. 6B, left; Step 1) (Siddiqui et al, 2019). Released KIF1C then recognizes the autoinhibited FHF complex via the interaction between Hook3$_{CC3}$ and KIF1C$_{H3BD}$ (Fig. 6B, left; Step 2) (Abid Ali et al, 2025). It is accompanied by the replacement of KIF1C-binding partner from PTPN21 with Hook3, as evidenced by our biochemical analyses (Figs. 1C,D and 6B, left; Step 2). Formation of the FHF − KIF1C complex (see Fig. 6A) augments the activation of both Hook3 and KIF1C, which is accompanied by disruption of intermolecular interaction among coiled coils of autoinhibited Hook3 (Fig. 6B, left; Step 3) (Abid Ali et al, 2025). Previous studies have shown that dynein and dynactin are then recruited to the amino-terminal domain of Hook3 undergoing activation, which synergistically

accelerates full-activation of the FHF complex and KIF1C-mediated anterograde cargo transport (Fig. 6B, left; Step 4) (Abid Ali et al, 2025).

As mentioned earlier, we verified that the three Hook3$_{CC3}$ − KIF1C$_{H3BD}$ binding-disrupting mutations (Y757A · F764A in KIF1C and V614E in Hook3; see Fig. 2) selected based on our crystal structure were effective in inhibiting interactions not only between Hook3 and KIF1C but also between KIF1C$_{H3BD}$ and PTPN21$_{FERM}$ (Fig. 3). Therefore, we considered that the introduction of these mutations would impair the release of KIF1C from the autoinhibited state and the consequent activation of Hook3 and KIF1C, which would interfere with KIF1C-dependent anterograde cargo transport (Fig. 6B, right). However, binding of the dynein and dynactin complex to the amino-terminal region of Hook3, which is necessary to facilitate minus-end-directed cargo transport, is not expected to be affected by the presence of the V614E mutation in Hook3. Thus, we hypothesized that dynein-dependent retrograde cargo transport would be dominant upon the expression of the Hook3$_{CC3}$ − KIF1C$_{H3BD}$ binding-disrupted mutant protein(s) (Fig. 6B, right). As expected, we detected enrichment of the EGFP-tagged mitochondria cargo in the perinuclear region of RPE1 cells expressing at least one of the two proteins in the binding-defective mutant form (Fig. 5B; Movies EV2–EV4).

## Concluding remark

Collectively, this study provides structural, biochemical, and cellular data demonstrating how Hook3, KIF1C, and PTPN21 interact with each other to control anterograde cargo transport. Our results, along with the data from previous studies, serve as a rational basis for generating a mechanistic model that answers several unresolved questions in the field: how KIF1C is fully activated by interacting with PTPN21 and Hook3, which target the same region of KIF1C; how the activation of Hook3 is modulated by KIF1C binding; and how the interaction between Hook3 and KIF1C leads to the assembly and activation of Hook3- and KIF1C-dependent cargo transport machinery.

## Methods

**Reagents and tools table**

| Reagent/resource | Reference or source | Identifier or catalog number |
|---|---|---|
| **Experimental models** | | |
| *E. coli* BL21(DE3) RIL | Novagen | Cat#69450-3CN |
| 293T | ATCC | CRL-3216 |
| hTERT RPE-1 | ATCC | CRL-4000 |

| Reagent/resource | Reference or source | Identifier or catalog number |
|---|---|---|
| **Recombinant DNA** | | |
| pBHA-Hook3 | Bioneer | N/A |
| pBHA-KIF1C | Bioneer | N/A |
| pET21a-His$_{10}$ − MBP −Hook3(553–590) | This paper | N/A |
| pET21a-His$_{10}$ − MBP −Hook3(553–624) | This paper | N/A |
| pET21a-His$_{10}$ − MBP −Hook3(553–624; V614E) | This paper | N/A |
| pET21a-His$_{10}$ − MBP −Hook3(553–718) | This paper | N/A |
| pET21a-His$_{10}$ − MBP − KIF1C(714–809) | This paper | N/A |
| pET21a-His$_{10}$ − MBP − KIF1C(714–809; Y757A and F764A) | This paper | N/A |
| pET21a-His$_{10}$ − MBP − PTPN21(19–308) | This paper | N/A |
| pKIF1C(1–1103)-3xFLAG−EGFP | This paper | N/A |
| pKIF1C(1–1103; Y757A and F764A)- 3xFLAG−EGFP | This paper | N/A |
| pKIF1C(1–1103)-3xFLAG −mTagBFP2 | This paper | N/A |
| pKIF1C(1–1103; Y757A and F764A)-3xFLAG−mTagBFP2 | This paper | N/A |
| pHook3(1–718)-Myc−FuRed | This paper | N/A |
| pHook3(1–718;V614E)-Myc −FuRed | This paper | N/A |
| pHook3(1–718;)-Myc−FuRed-FRB | This paper | N/A |
| pHook3(1–718; V614E)-Myc −FuRed-FRB | This paper | N/A |
| pFKBP-EGFP−MoA | This paper | N/A |
| **Antibodies** | | |
| Anti-Flag M2 antibody | Sigma-Aldrich | Cat#F1804 |
| Protein A/G PLUS-Agarose | Santa Cruz | Cat#sc-2003 |
| DYKDDDDK Tag (D6W5B) Rabbit mAb | Cell Signaling | Cat#14793 |
| Anti-Myc tag antibody | Abcam | Cat#ab9106 |
| GAPDH Loading Control Monoclonal Antibody (GA1R) | Invitrogen | Cat# MA5-15738 |
| IRDye® 680RD Goat anti-Rabbit IgG Secondary Antibody | LI-COR | Cat# 926-68071 |
| IRDye® 800CW Goat anti-Mouse IgG Secondary Antibody | LI-COR | Cat# 926-32210 |
| **Chemicals, enzymes and other reagents** | | |
| Luria-Bertani media | Condalab | Cat#1551.00 |
| Isopropyl β-ᴅ-1-thiogalactopyranoside | Goldbio | Cat#367-93-1 |
| L-Selenomethionine | TCI | Cat#S0442 |

| Reagent/resource | Reference or source | Identifier or catalog number |
|---|---|---|
| Tris(hydroxymethyl) aminomethane | Duchefa | Cat#T1501.5000 |
| Sodium chloride | Junsei | Cat#19015S1250 |
| β-mercaptoethanol | Merck | Cat#M6250 |
| Polyethylene glycol 4000 | Merck | Cat#1546569 |
| Bis-Tris propane | Merck | Cat#B6755 |
| Isopropanol | Merck | Cat#1096341011 |
| 100% Tacsimate pH 7.0 | Hampton Research | Cat#50-258-5 |
| Glycerol | Junsei | Cat#27210S0350 |
| Hydrochloric acid | Junsei | Cat#20010S0350 |
| jetPRIME | Polyplus | Cat#101000046 |
| Pierce™ IP lysis buffer | Thermo Scientific | Cat#87787 |
| Halt™ protease inhibitor cocktail | Thermo Scientific | Cat#87786 |
| Rapamycin | LC Laboratories | Cat#R-5000 |
| **Software** | | |
| *HKL*2000 | Otwinowski and Minor, 1997 | https://www.hkl-xray.com/hkl-2000 |
| PHENIX | Adams et al, 2010 | https://www.phenix-online.org |
| COOT | Emsley and Cowtan, 2004 | https://www2.mrc-lmb.cam.ac.uk/personal/pemsley/coot/ |
| Phaser | McCoy et al, 2007 | |
| Nikon imaging software (NIS-elements AR analysis 64-bit version 5.21) | Nikon | |
| **Other** | | |
| Amicon Ultra centrifugal filter | Millipore | Cat#UFC901096 |
| Ni-NTA Sepharose resin | Cytiva | Cat#17526802 |
| HiLoad 26/600 Superdex 75 prep grade column | Cytiva | Cat#28-9893-34 |
| Superdex 200 Increase 10/300 GL column | Cytiva | Cat#28-9909-44 |

## Preparation of recombinant proteins

DNA fragments encoding Hook3(553–718/553–624/553–590), KIF1C(714–809), and PTPN21(19–308) were amplified by polymerase chain reaction (PCR) and inserted into the pET21a vector (Novagen), which was modified to produce a protein amino-terminally tagged with a (His)$_{10}$-linked maltose binding protein (MBP). These constructs were used as templates for the preparation of mutant proteins Hook3(553–624; V614E) and KIF1C(714–809; Y757A and F764A). The cloned plasmids were

transformed into *E. coli* BL21(DE3) RIL strain (Novagen). The transformed cells were inoculated in Luria-Bertani media and incubated at 37 °C until they reached an $OD_{600}$ of ~0.5. The protein expression was induced by adding 0.2 mM isopropyl β-D-1-thiogalactopyranoside for 16 h at 18 °C. The expressed proteins were purified using a Ni-NTA affinity chromatography column (Cytiva) and a HiLoad® 26/600 Superdex® 75 prep grade size-exclusion chromatography column (Cytiva). The $His_{10}$-linked maltose-binding protein tag was digested using tobacco etch virus protease treatment and then removed using an additional Ni-NTA affinity purification step. After purification, the protein samples were stored in a buffer containing 50 mM Tris-HCl (pH 7.5), 500 mM NaCl, 10% glycerol, and 3 mM β-mercaptoethanol at −80 °C. Selenomethionine (SeMet)-substituted Hook3(553–624) was produced using the *E. coli* BL21(DE3) RIL strain inoculated in M9 minimal medium, and purified in the same manner as the native protein.

## Crystallization and structural determination of Hook3 alone or in a complex with KIF1C

Crystallization was performed using native Hook3(553–624) in the KIF1C(714–809)-bound form and SeMet-substituted apo Hook3(553–624), which were concentrated to 30 and 20 mg/mL, respectively. All the crystals were produced using the sitting drop diffusion method at 4 °C with drops composed of 0.2 μL protein and 0.2 μL reservoir solution: 0.1 M sodium citrate/citric acid (pH 5.5), 20% (w/v) polyethylene glycol 4000, and 20% (v/v) isopropanol for the Hook3(553–624) − KIF1C(714–809) complex; 0.1 M Bis-Tris propane (pH 6.8) and 60% Tacsimate (pH 7.0) for SeMet-substituted Hook3(553–624). The crystals were soaked in a cryoprotectant solution containing the mother liquor supplemented with 15% glycerol as the cryoprotectant for the Hook3 − KIF1C complex. The crystals were used to collect diffraction data on beamlines 11C (for the Hook3 − KIF1C complex) and 5 C (for native Hook3) at the Pohang Accelerator Laboratory, Korea. The diffraction data were scaled using the program *HKL*2000 (Otwinowski and Minor, 1997). The structure of SeMet-substituted Hook3 was determined by single-wavelength anomalous diffraction using the program AutoSol (Zwart et al, 2008), which served as a search model for the molecular replacement method used for the structure determination of native Hook3 and Hook3 − KIF1C complex with the program Phaser (McCoy et al, 2007). The programs PHENIX (Adams et al, 2010) and Coot (Emsley and Cowtan, 2004) were used for model building and refinement. The crystallographic data are presented in Table EV1.

## ITC

$His_{10}$ − MBP−Hook3(553–624), $His_{10}$ − MBP − PTPN21(19–308), and $His_{10}$ − MBP − KIF1C(714–809) were purified and dialyzed in a buffer containing 50 mM Tris-HCl (pH 7.5), 500 mM NaCl, and 3 mM β-mercaptoethanol. All samples were degassed for 20 min using a vacuum system before the experiments. ITC was performed by titrating 0.8 mM $His_{10}$ − MBP−Hook3(553–624) or $His_{10}$ − MBP − PTPN21(19–308) into 0.04 mM $His_{10}$ − MBP − KIF1C(714–809). The ITC measurements were performed using a MicroCal VP-ITC microcalorimeter system at 25 °C, and analyzed using Origin® software (OriginLab Co.).

## Native gel electrophoresis

Purified $His_{10}$ − MBP− Hook3(553–718/553–624/553–590) proteins were incubated for 16 h at 4 °C either individually or in combination with $His_{10}$ − MBP − KIF1C(714–809) at a concentration of 10 μM. The samples were then subjected to 10% native gel and analyzed by electrophoresis at 120 V for 2 h on ice.

## Protein replacement analysis using size-exclusion chromatography

The six prepared protein samples, including Hook3(553–624), PTPN21(19–308), $His_{10}$ − MBP − KIF1C(714–809) bound to Hook3(553–624) called [KIF1C + Hook3], $His_{10}$ − MBP − KIF1C(714–809) bound to PTPN21(19–308) called [KIF1C + PTPN21], [KIF1C + Hook3] mixed with PTPN21, and [KIF1C + PTPN21] mixed with Hook3, were subjected to a Superdex™ 200 Increase 10/300 GL column. The size-exclusion chromatography analysis was performed with 12 μM protein samples in a buffer containing 50 mM Tris-HCl (pH 7.5), 500 mM NaCl, 10% glycerol, and 3 mM β-mercaptoethanol.

## Plasmid preparation for cellular assays

DNA fragments encoding KIF1C−Flag and Hook3−Myc were amplified by PCR. The KIF1C−Flag fragment was inserted into pEGFP-N1 (Clontech) to produce the KIF1C−Flag−EGFP construct. The Hook3−Myc fragment was inserted into a modified pEGFP-N1 vector, which tagged the protein with carboxyl-terminal FuRed instead of EGFP, to produce the Hook3−Myc−FuRed construct. To tag mitochondria with EGFP, PCR-amplified DNA fragments for FKBP and residues 490–527 of MoA were sequentially inserted into pEGFP-C1 (Clontech) using the NheI and AgeI for FKBP and BsrGI and BamHI for MoA, resulting in the preparation of the FKBP − EGFP−MoA construct. For tracking the mitochondria cargo transport, EGFP in the KIF1C−Flag−EGFP construct was replaced with mTagBFP2 using AgeI and BsrGI (KIF1C−Flag−mTagBFP2) and the PCR-amplified DNA fragment encoding FRB was inserted into the Hook3−Myc−FuRed construct using BsrGI (Hook3−Myc−FuRed−FRB). The reactions were performed using Gibson Assembly Cloning Kit (NEB). These constructs were used as templates for the preparation of Hook3(V614E) and KIF1C(Y757A and F764A) mutant.

## Co-immunoprecipitation assay and western blotting

HEK293T cells (ATCC), which were routinely tested for mycoplasma during maintenance, were grown in 10% fetal bovine serum (Invitrogen)-supplemented Dulbecco's Modified Eagle's Medium (Gibco) at 37 °C in a humidified 10% $CO_2$ atmosphere. The cells were transfected with the vectors encoding KIF1C − FLAG − EGFP or Hook3−Myc−FuRed using jetPrime® (Polyplus). The cells were harvested 16 h post transfection and cell extracts were prepared using Pierce™ IP Lysis Buffer (Thermo Scientific) containing Halt™ Protease Inhibitor Cocktail (Thermo Scientific). The lysates were incubated with a monoclonal anti-Flag M2 antibody (Sigma) overnight at 4 °C on a rocking platform. Protein A/G Plus-Agarose (Santa Cruz Biotechnology) was added to the mixture and incubated for 2 h at 4 °C while rotating. The beads were collected

by centrifugation, washed in lysis buffer, and resuspended in sodium dodecyl sulfate gel loading buffer. The proteins were separated on Bolt™ 4–12% Bis-Tris Plus gels and transferred to a polyvinylidene difluoride membrane using iBlot™ Transfer Stack (Gibco). For immunoblotting, the membranes were blocked with Odyssey™ TBS Blocking Buffer and incubated with rabbit anti-Flag (Cell Signaling Technology), rabbit anti-Myc (Abcam), and mouse anti-GAPDH (Thermo Scientific) antibodies overnight at 4 °C with shaking. After washing three times with Tris-buffered saline with 0.1% Tween® 20, the membranes were incubated with IRDye 680RD Goat anti-Rabbit IgG (H + L) and IRDye 800CW Goat anti-Mouse IgG (H + L) (LI-COR) for 1 h at room temperature. After washing three times again, the membrane was imaged using LI-COR Odyssey™ imaging system and analyzed using the Image Studio™ software.

### Protein distribution and cargo transport analysis

hTERT-RPE1 cells (ATCC), which were routinely tested for mycoplasma during maintenance, were cultured in 10% fetal bovine serum-supplemented Dulbecco's Modified Eagle Medium/ Nutrient Mixture F-12 (Gibco), at 37 °C in a humidified 10% $CO_2$ atmosphere. To analyze the subcellular distribution of Hook3 and KIF1C, RPE1 cells were transfected with vectors encoding KIF1C − FLAG − EGFP or Hook3−Myc−FuRed. To track the mitochondrial cargo transport, RPE1 cells were transfected with vectors encoding KIF1C−mTagBFP2, Hook3−FuRed−FRB, or FKBP − EGFP−MoA. Transfection was performed using an electroporation system (Neon Transfection System; Invitrogen) according to the manufacturer's instructions, under optimized conditions (three pulses of 1350 V for 10 ms) for high transfection efficiency. The cells were then plated on fibronectin-coated 96-well plates (μ-Plate 96 Well 3D ibiTreat; ibidi GmbH). Live-cell imaging was performed using a Nikon A1R confocal microscope (Nikon Instruments) mounted on a Nikon Eclipse Ti body equipped with a Nikon CFI Plan Apochromat VC objective (60×/1.4 numerical aperture; Nikon Instruments). A Chamlide TC system placed on a Nikon A1R confocal microscope stage was used to maintain environmental conditions at 37 °C and 10% $CO_2$ (Live Cell Instruments Inc.). All images were obtained 16 h after transfection and analyzed using the Nikon imaging software (NIS-elements AR analysis 64-bit version 5.21; Laboratory Imaging). To analyze tail enrichment, the mean intensity was measured from the tip of the cell tail to the perinuclear area, and the ratio of tail/cytoplasmic intensity was calculated. To analyze the mitochondrial transport, RPE1 cells expressing all three fluorescent proteins were selected. Images were acquired every 1 min from −2 to 16 min after treatment with 500 nM rapamycin. To determine the distribution of tag-conjugated mitochondria, mean intensity of region of interest referred from accumulation of KIF1C −mTagBFP2 at the cell tip was measured, and the mean intensity of the perinuclear area was measured independently for cytoplasmic intensity. The ratio of tail-to-cytoplasm intensity was then calculated. To quantify mitochondrial redistribution, line profiles from the center of the cells to the tips were acquired at each time point. The mean intensity of each pixel on one-line profile was normalized to the maximum value of the pixel intensity. The normalized intensity (0.0–1.0) was plotted against time after rapamycin treatment.

## Data availability

Structure factors and coordinates have been deposited in the Protein Data Bank (https://www.rcsb.org/) with entry numbers 9KO8 for Hook3(553–624) − KIF1C(714–809) (https://www.rcsb.org/structure/9KO8) and 9KNS for Hook3(553–624) (https://www.rcsb.org/structure/9KNS).

The source data of this paper are collected in the following database record: biostudies:S-SCDT-10_1038-S44319-025-00458-w.

## Peer review information

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

## Acknowledgements

The beamlines 5C and 11C at the Pohang Accelerator Laboratory in Korea were used in this study. We are grateful to Dr. Myung Hee Kim, Dr. Jungwon Hwang (Korea Research Institute of Bioscience and Biotechnology, Korea) and Dr. Eunha Hwang (Korea Basic Science Institute, Korea) for assistance with ITC experiments. This work was supported by the National Research Council of Science and Technology (CRC22021-700), the KRIBB Research Initiative Program (KGM9952522), and the National Research Foundation of Korea (RS-2023-00278696 and NRF-2022R1C1C200623113), which were funded by the Ministry of Science and ICT (MSIT) of the Republic of Korea.

## Author contributions

**Hye Seon Lee**: Conceptualization; Investigation; Writing—original draft; Writing—review and editing. **Daseuli Yu**: Conceptualization; Funding acquisition; Investigation; Writing—original draft. **Kyoung Eun Baek**: Investigation. **Ho-Chul Shin**: Methodology. **Seung Jun Kim**: Supervision; Funding acquisition. **Won Do Heo**: Supervision. **Bonsu Ku**: Conceptualization; Supervision; Funding acquisition; Investigation; Writing—original draft; Writing—review and editing.

Source data underlying figure panels in this paper may have individual authorship assigned. Where available, figure panel/source data authorship is listed in the following database record: biostudies:S-SCDT-10_1038-S44319-025-00458-w.

## Disclosure and competing interests statement

The authors declare no competing interests.

# Expanded View Figures

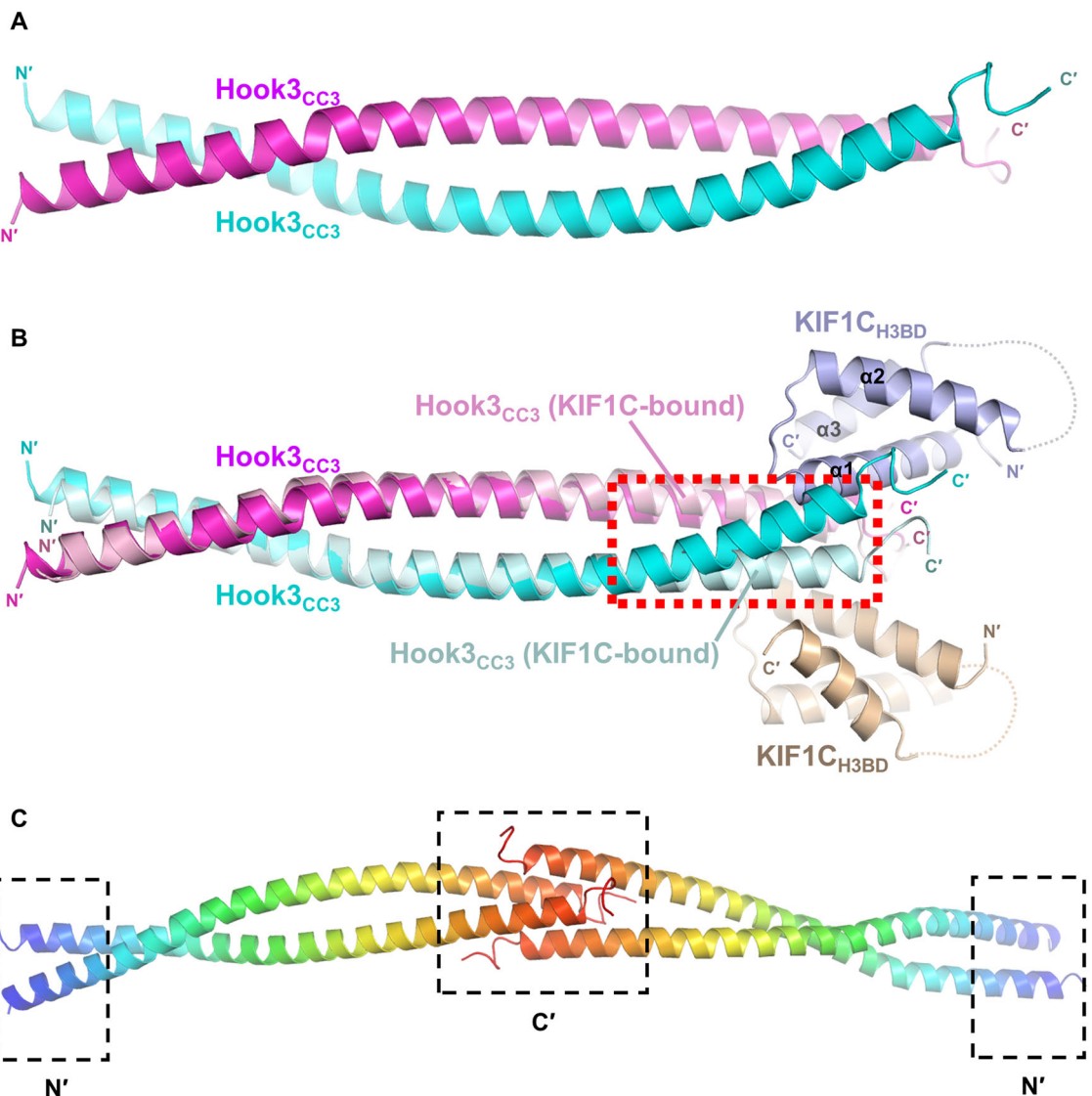

**Figure EV1. Crystal structure of Hook3(553–624).**

(**A**) Crystal structure of apo Hook3(553–624; magenta and cyan) shown as ribbon representation. The amino- and carboxyl-termini of each polypeptide are indicated as N′ and C′, respectively. (**B**) Cartoon representation of superposed structures of Hook3(553–624) in the apo form (magenta and cyan) and in KIF1C(714–809; navy and beige)-bound form (pink and mint). Dashed lines linking α2 and α3 of KIF1C represent invisible regions in the crystal structure owing to poor electron density. (**C**) Crystal packing of apo Hook3(553–624). Two symmetry mates of apo Hook3(553–624) are shown as ribbon representation. The four Hook3 molecules are gradually colored from blue (N′; residue 553) to orange (C′; residue 624).

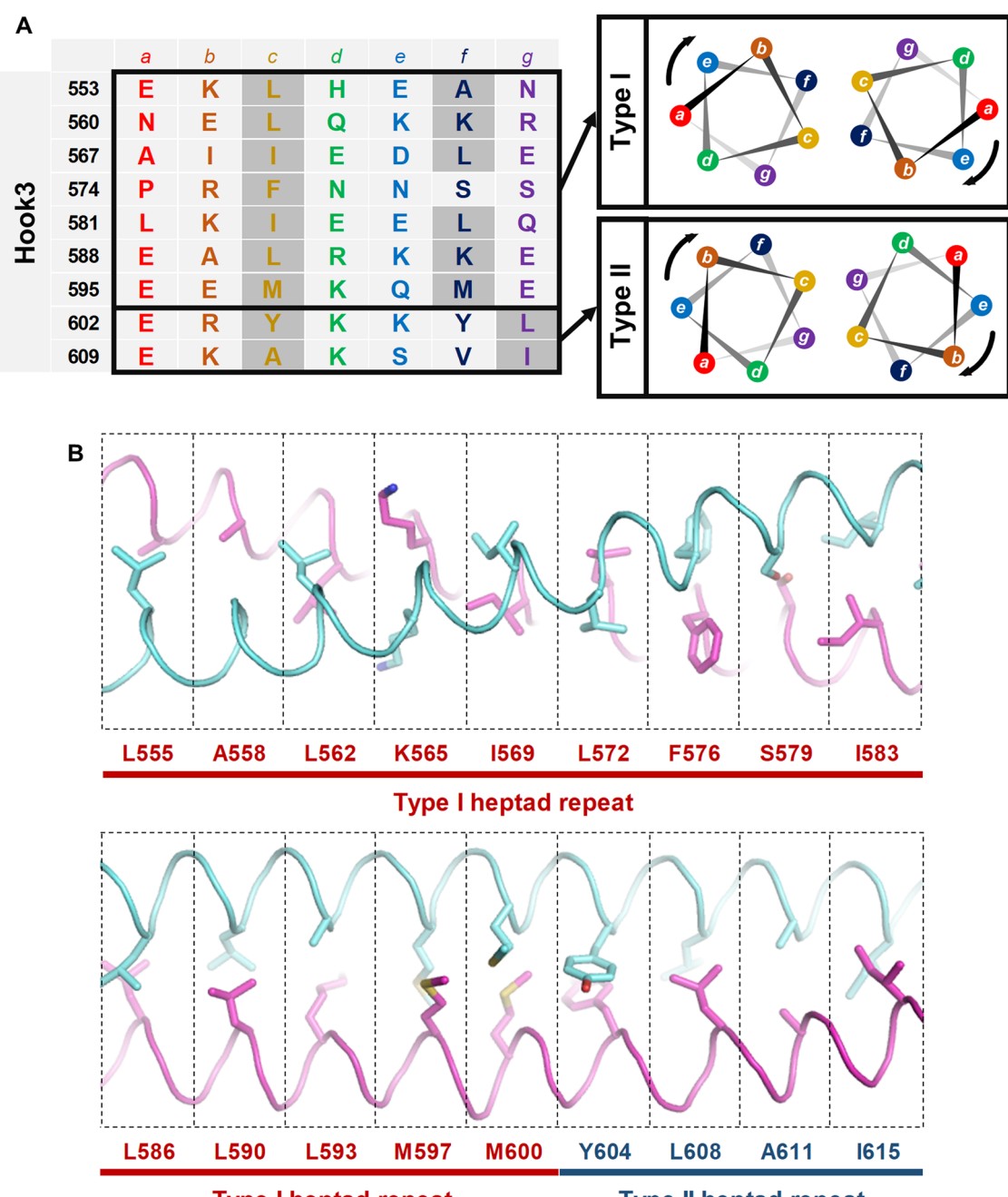

**Figure EV2. Coiled-coil conformation of Hook3.**

(A) Two heptad repeat types present in Hook3(553–624). (Left) Hook3 residues are assigned by the heptad repeat position (*a–g*). Hydrophobic residues involved in the intermolecular interaction for the coiled-coil formation are shaded in gray. (Right) Helical wheel representations. Black boxes indicate two different heptad repeat types in Hook3. (B) Dimeric interface of the Hook3 coiled coils is shown in pink and cyan. The key residues for the coiled-coil formation are represented as sticks with labels shown at the bottom (red, type I; blue, type II heptad repeat).

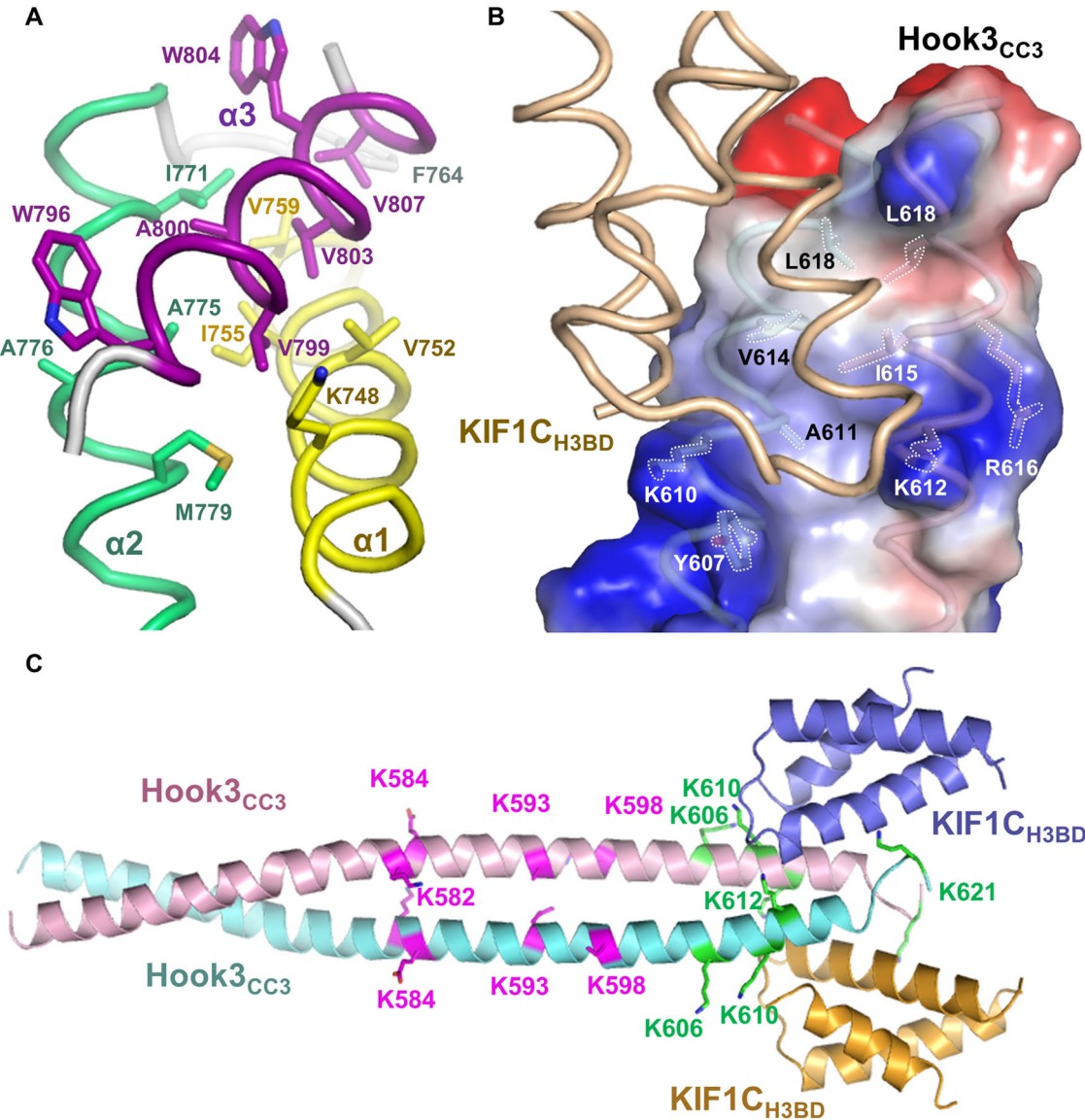

**Figure EV3. Detailed structural analysis of the Hook3 — KIF1C complex.**

(A) Intramolecular hydrophobic interaction in KIF1C(714–809). Loops are indicated in gray and the α1, α2, and α3 helices are represented in yellow, green, and purple, respectively. Hydrophobic residues involved in protein folding are represented as sticks and labeled. (B) Hook3(553–624) homodimer is shown as an electrostatic surface representation together with bound KIF1C(714–809) colored in beige. The Hook3 residues (black, from one Hook3 protomer; white, from another molecule) that constitute a large extended hydrophobic surface are shown in sticks with labels. (C) Lysine residues of Hook3 previously reported to be cross-linked with those of KIF1C (Abid Ali et al, 2025; Data ref: Abid Ali et al, 2025) are represented on our Hook3 — KIF1C complex structure as sticks with labels. Among them, those in the proximity of the complex interface are shown in green, whereas the rest are represented in magenta.

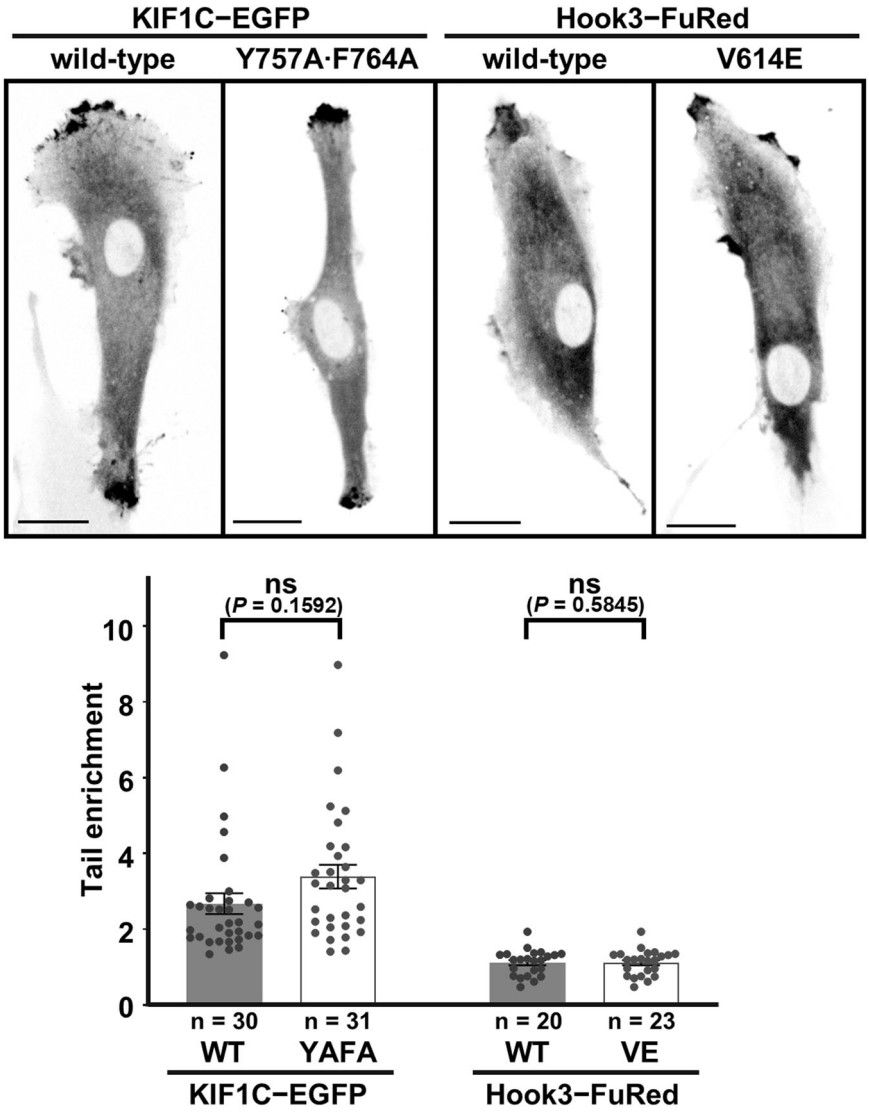

**Figure EV4. Intracellular localization of Hook3 and KIF1C expressed separately.**

(Top) Fluorescent images of RPE1 cells expressing KIF1C — EGFP or Hook3—FuRed at 16 h post transfection. Scale bars, 10 μm. (Bottom) Tail enrichment of KIF1C — EGFP and HooK3—FuRed in RPE1 cells. Tail/cytoplasm ratios of each protein were analyzed. Values are means ± standard error of the mean. ns, not significant by the Student's two-tailed *t* test. WT wild-type, YAFA Y757A · F764A, VE V614E.

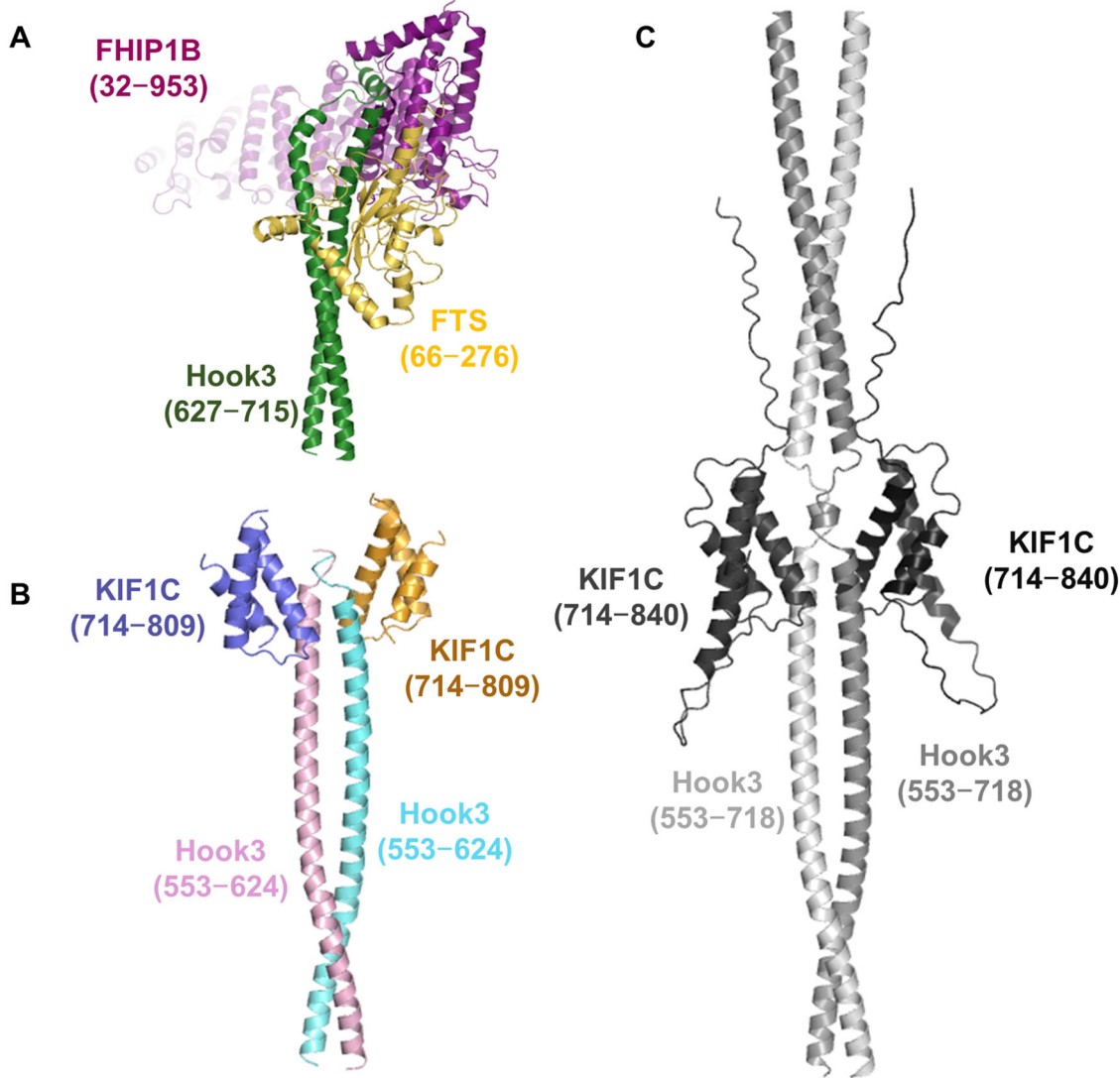

**Figure EV5. Experimentally determined and AlphaFold2-predicted structures of Hook3-containing complexes.**

(A) Cryo-EM structure of the FTS(66–276)—Hook3(627–715) — FHIP1B(32–953) complex (PDB code: 8QAT, Abid Ali et al, 2025; Data ref: Abid Ali et al, 2025). (B) Crystal structure of the Hook3(553–624) — KIF1C(714–809) complex (PDB code: 9KO8). (C) AlphaFold2-based structural model of the Hook3(553–718) — KIF1C(714–840) complex.

