## [Peer Review File · EMBO Reports]

Molecular basis for assembly and activation of the Hook3–KIF1C complex-dependent transport machinery

Hye Seon Lee, Daseuli Yu, Kyoung Eun Baek, Ho-Chul Shin, Seung Jun Kim, Won Do Heo, and Bonsu Ku

Corresponding author(s): Bonsu Ku (bku@kribb.re.kr) , Seung Jun Kim (ksj@kribb.re.kr), Won Do Heo (wondo@kaist.ac.kr)

Review Timeline:

Submission Date:	3rd Dec 24
Editorial Decision:	28th Jan 25
Revision Received:	25th Feb 25
Editorial Decision:	21st Mar 25
Revision Received:	27th Mar 25
Accepted:	3rd Apr 25

Editor: Deniz Senyilmaz Tiebe

Transaction Report:

Dear Dr. Ku,

Thank you for submitting your research manuscript to our journal, which was now seen by two referees, whose reports are copied below.

Referees express interest in the proposed mechanism for assembly and activation of the Hook3–KIF1C complex-dependent transport machinery. However, they also raise some concerns that need to be addressed to consider publication here.

Given these positive recommendations, we would like to invite you to revise your manuscript with the understanding that the referee concerns (as in their reports) must be fully addressed and their suggestions taken on board. Please address all referee concerns in a complete point-by-point response. Acceptance of the manuscript will depend on a positive outcome of a second round of review. It is EMBO reports policy to allow a single round of major experimental revision only and acceptance or rejection of the manuscript will therefore depend on the completeness of your responses included in the next, final version of the manuscript.

We realize that it is difficult to revise to a specific deadline. In the interest of protecting the conceptual advance provided by the work, we recommend a revision within 3 months. Please discuss the revision progress ahead of this time with me if you require more time to complete the revisions, or if you have questions or comments regarding the revision (also by video chat).

1. A data availability section providing access to data deposited in public databases is missing (where applicable).
2. Your manuscript contains statistics and error bars based on $n=2$. Please use scatter plots in these cases.

You can submit the revision either as a Scientific Report or as a Research Article. For Scientific Reports, the revised manuscript can contain up to 5 main figures and 5 Expanded View figures, and it should not exceed 27000 characters. If the revision leads to a manuscript with more than 5 main figures it will be published as a Research Article. In this case the Results and Discussion section should be separate. If a Scientific Report is submitted, these sections have to be combined. This will help to shorten the manuscript text by eliminating some redundancy that is inevitable when discussing the same experiments twice. In either case, all materials and methods should be included in the main manuscript file.

<<https://www.embopress.org/page/journal/14693178/authorguide#expandedview>>

4) a .docx formatted letter INCLUDING the reviewers' reports and your detailed point-by-point responses to their comments. As part of the EMBO publication's Transparent Editorial Process, EMBO reports publishes online a Review Process File (RPF) to accompany accepted manuscripts. This File will be published in conjunction with your paper and will include the referee reports, your point-by-point response and all pertinent correspondence relating to the manuscript.

<https://www.embopress.org/page/journal/14693178/authorguide#transparentprocess>

You are able to opt out of this by letting the editorial office know (emboreports@embo.org). If you do opt out, the Review Process File link will point to the following statement: "No Review Process File is available with this article, as the authors have

chosen not to make the review process public in this case."

5) a complete author checklist, which you can download from our author guidelines

<https://www.embopress.org/page/journal/14693178/authorguide>. Please insert information in the checklist that is also reflected in the manuscript. The completed author checklist will also be part of the RPF.

6) Please note that all corresponding authors are required to supply an ORCID ID for their name upon submission of a revised manuscript (<<https://orcid.org/>>). Please find instructions on how to link your ORCID ID to your account in our manuscript tracking system in our Author guidelines

<<https://www.embopress.org/page/journal/14693178/authorguide#authorshipguidelines>>

7) Before submitting your revision, primary datasets produced in this study need to be deposited in an appropriate public database (see <https://www.embopress.org/page/journal/14693178/authorguide#datadeposition>). Please remember to provide a reviewer password if the datasets are not yet public. The accession numbers and database should be listed in a formal "Data Availability" section placed after Materials & Method (see also

<https://www.embopress.org/page/journal/14693178/authorguide#datadeposition>). Please note that the Data Availability Section is restricted to new primary data that are part of this study. * Note - All links should resolve to a page where the data can be accessed. *

Additional information on source data and instruction on how to label the files are available:

<https://www.embopress.org/page/journal/14693178/authorguide#sourcedata>

9) Our journal encourages inclusion of *data citations in the reference list* to directly cite datasets that were re-used and obtained from public databases. Data citations in the article text are distinct from normal bibliographical citations and should directly link to the database records from which the data can be accessed. In the main text, data citations are formatted as follows: "Data ref: Smith et al, 2001" or "Data ref: NCBI Sequence Read Archive PRJNA342805, 2017". In the Reference list, data citations must be labeled with "[DATASET]". A data reference must provide the database name, accession number/identifiers and a resolvable link to the landing page from which the data can be accessed at the end of the reference. Further instructions are available at <http://www.embopress.org/page/journal/14693178/authorguide#referencesformat>

10) Regarding data quantification (see Figure Legends:

<https://www.embopress.org/page/journal/14693178/authorguide#figureformat>)

12) Please also note our reference format:

13) All Materials and Methods need to be described in the main text using our 'Structured Methods' format, which is required for all research articles. According to this format, the Methods section includes a Reagents and Tools Table (listing key reagents, experimental models, software and relevant equipment and including their sources and relevant identifiers) followed by a Methods and Protocols section describing the methods using a step-by-step protocol format. The aim is to facilitate adoption of the methodologies across labs. More information on how to adhere to this format as well as a downloadable template (.docx) for the Reagents and Tools Table can be found in our author guidelines:
<https://www.embopress.org/page/journal/14693178/authorguide#structuredmethods>.

An example of a Method paper with Structured Methods can be found here:
<https://www.embopress.org/doi/10.15252/msb.20178071>.

I look forward to seeing a revised version of your manuscript when it is ready. Please let me know if you have questions or comments regarding the revision.

Kind regards,

Deniz Senyilmaz Tiebe

Deniz Senyilmaz Tiebe, PhD
Scientific Editor
EMBO Reports

Referee #1:

This paper offers important insights into the interaction between KIF1C, Hook3, and PTPN21, and reveals that Hook3 and PTPN21 competitively bind for KIF1C. Authors report a high-resolution structural model of Hook3 forming a 2:2 heterotetrameric complex with KIF1C using X-ray crystallography and analyze the intermolecular association between Hook3 and KIF1C. Depending on the structural-based mutagenesis, the authors claim that expressing both proteins leads to colocalization between KIF1C and Hook3, which is disrupted by mutant proteins are expressed. The data in this paper using structural, biochemical and cell biology techniques offers a novel model for how the Hook3-KIF1C complex assembles and activates.

The authors point out that their structure helps to resolve conflicting data that maps the interaction between HOOK3 and KIF1C, establishing the mode of interaction between the two proteins. We only have minor comments related to this work.

We only suggest text changes with the following typos identified:

-In the results, 'In contrast, no complex formation was detected in lanes 6-8 which contained at least on e mutant protein (Figure 3A).'

-In the results, 'and therefore they were used for tracking KIF1C-mediated subcellular transport in previous18,24,28 and in this study" should be "and therefore they were used for tracking KIF1C-mediated subcellular transport in previous studies18,24,28 and this study"

-In the discussion part, ' Moreover, tail enrichment of Hook3 and KIF1C (Figure 4) and transport of the Hook3-conjugated mitochondria cargo to the cell tips (Figure 5) occurred only when the two proteins were co-expressed as wild-type, but not when either one of them was expressed in the binding-disrupted mutant form. '.

Referee #2:

This study by Lee et al. examined the structural and functional basis of the interaction between cargo-adaptor protein Hook3 and the kinesin motor KIF1C, two proteins essential for microtubule-associated cargo transport. Through a combination of biochemical, structural, and cellular experiments, the researchers determined that Hook3 forms a 2:2 heterotetrameric complex with KIF1C, where two KIF1C molecules bind to a Hook3 homodimer. They used native gel electrophoresis to verify the direct interaction between Hook3 and KIF1C, identifying the KIF1C-binding region in Hook3. Isothermal titration calorimetry (ITC) was used to quantify the binding affinities of KIF1C with Hook3 and PTPN21. The crystal structure of the Hook3 (553-624) complexed with KIF1C (714-809) was determined, revealing that the interaction is mainly mediated by hydrophobic residues and that the α 1 helix and α 1- α 2 loop of KIF1C are crucial for binding to Hook3. Site-directed mutagenesis, followed by native gel

electrophoresis and co-immunoprecipitation assays, confirmed that specific residues in both Hook3 and KIF1C are critical for complex formation. Furthermore, using a chemically inducible dimerization system and live-cell imaging, the study demonstrated that the Hook3-KIF1C interaction is essential for the anterograde transport of mitochondria in RPE1 cells. These findings provide a detailed structural and mechanistic understanding of how Hook3 and KIF1C cooperate to facilitate cargo transport. This research is important because it provides a more detailed understanding of the molecular mechanisms underlying microtubule-based cargo transport, a fundamental process in cell biology that is crucial for various cellular functions such as cell development, polarity, morphology and migration. Thus, this study should be interesting to a wide audience of cell biologists, biochemists, and structural biologists. I found the results compelling, and they add to a growing body of knowledge towards understanding the intricate mechanisms of control for molecular motor-driven transport in cells. The paper complements a recently published study on these proteins by Ali et al., which is discussed here. I support publication after the authors address my comments below.

Specific comments:

1. It is unclear to me what the "microtubule-anchoring Hook domain" means. The phrase implies that the Hook domain binds directly to microtubules, but I am unaware of any evidence of this. Please clarify the meaning of this terminology.
2. Figure 1 would be much easier to understand if it included a diagram of each full-length protein, highlighting the fragments/domains used in each experiment (similar to the diagram in Fig. 2).
3. "we found that EGFP-tagged mitochondria were enriched in the tail region (Figure 5B, marked by dashed rectangles". I do not see dashed rectangles in this figure panel. Perhaps the authors meant colored rectangles?
4. The authors use "type I" and "type II" heptad repeat to describe regions of Hook3 but these terms are not standard in the field that I am aware of. The authors may want to be clear that they are defining these terms in this work.
5. The implication of the competitive interaction between PTPN21 and Hook3 for KIF1C could be more fully elucidated in the discussion. Does the phosphatase activity of PTPN21 play any role?
6. Fig. 6B is somewhat complicated. There is no step showing the competitive interaction between PTPN21 and Hook3 for KIF1C. Is this assumed to happen between step 1 and step 2? If so, this should probably be labeled. It is unclear to me what "normal" and "depletion" interactions mean in the labeling. I don't fully understand the right side of the cartoon.

Editor's comments

1. A data availability section providing access to data deposited in public databases is missing (where applicable).

→ The data availability section is now added in the manuscript during revision.

2. Your manuscript contains statistics and error bars based on $n=2$. Please use scatter plots in these cases.

→ Our manuscript does not contain any statistics and error bars based on $n=2$: The numbers of conducted experiments are [30, 31, 20, 23 (for Figure S4)], [25, 25, 12, 19 (for Figure 4B)], [24, 25, 12, 19 (for Figure 4C)], [29, 27, 24, 23, 27, 24, 25, 21 (for Figure 5D)], and [21, 7, 10, 9 (for Figure 5E)].

→ Our revised manuscript including legends for the main and EV figures highlighted the revision changes in red.

→ We provided all figures in accordance with EMBO guidelines.

3) We replaced Supplementary Information with Expanded View (EV) Figures and Tables that are collapsible/expandable online. A maximum of 5 EV Figures can be typeset. EV Figures should be cited as 'Figure EV1, Figure EV2' etc... in the text and their respective legends should be included in the main text after the legends of regular figures.

- For the figures that you do NOT wish to display as Expanded View figures, they should be bundled together with their legends in a single PDF file called *Appendix*, which should start with a short Table of Content. Appendix figures should be referred to in the main text as: "Appendix Figure S1, Appendix Figure S2" etc. See detailed instructions regarding expanded view here: <<https://www.embopress.org/page/journal/14693178/authorguide#expandedview>>;

➔ Our manuscript contains six main figures and five EV figures. We also prepared an appendix PDF file for Appendix Table S1.

4) a .docx formatted letter INCLUDING the reviewers' reports and your detailed point-by-point responses to their comments. As part of the EMBO publication's Transparent Editorial Process, EMBO reports publishes online a Review Process File (RPF) to accompany accepted manuscripts. This File will be published in conjunction with your paper and will include the referee reports, your point-by-point response and all pertinent correspondence relating to the manuscript.

<https://www.embopress.org/page/journal/14693178/authorguide#transparentprocess>

➔ We prepared the Response_to_Reviewers as a separate file.

5) a complete author checklist, which you can download from our author guidelines <https://www.embopress.org/page/journal/14693178/authorguide>. Please insert information in the checklist that is also reflected in the manuscript. The completed author checklist will also be part of the RPF.

➔ We checked it.

6) Please note that all corresponding authors are required to supply an ORCID ID for their name upon submission of a revised manuscript (<<https://orcid.org/>>). Please find instructions on how to link your ORCID ID to your account in our manuscript tracking system in our Author guidelines

<https://www.embopress.org/page/journal/14693178/authorguide#authorshipguidelines>

➔ We added the ORCID IDs in the title page of the revised manuscript.

➔ We deposited the two crystal structures in the Protein Data Bank (PDB) database, which were release to the public at February 26th, 2025.

➔ We provided the source data as requested by EMBO press.

9) Our journal encourages inclusion of *data citations in the reference list* to directly cite datasets that were re-used and obtained from public databases. Data citations in the article text are distinct from normal bibliographical citations and should directly link to the database records from which the data can be accessed. In the main text, data citations are formatted as follows: "Data ref: Smith et al, 2001" or "Data ref: NCBI Sequence Read Archive PRJNA342805, 2017". In the Reference list, data citations must be labeled with "[DATASET]". A data reference must provide the database name, accession number/identifiers and a resolvable link to the landing page from which the data can be accessed at the end of the reference. Further instructions are available at <http://www.embopress.org/page/journal/14693178/authorguide#referencesformat>

➔ We labeled the cited data as "Data ref:" in the revised manuscript and [DATA SET] in the reference list.

➔ We checked all the notifications.

11) The journal requires a statement specifying whether or not authors have competing interests (defined as all potential or actual interests that could be perceived to influence the presentation or interpretation of an article). In case of competing interests, this must be specified in your disclosure statement. Further information:

<https://www.embopress.org/competing-interests>

➔ We declared that no competing interests exist

12) Please also note our reference format:

➔ We checked the reference format.

13) All Materials and Methods need to be described in the main text using our 'Structured Methods' format, which is required for all research articles. According to this format, the Methods section includes a Reagents and Tools Table (listing key reagents, experimental models, software and relevant equipment and including their sources and relevant identifiers) followed by a Methods and Protocols section describing the methods using a step-by-step protocol format. The aim is to facilitate adoption of the methodologies across labs. More

information on how to adhere to this format as well as a downloadable template (.docx) for the Reagents and Tools Table can be found in our author guidelines: <https://www.embopress.org/page/journal/14693178/authorguide#structuredmethods>.

➔ We revised the Materials and Methods section in accordance with above notification.

Reviewers' comments

> Reviewer 1

1. We only suggest text changes with the following typos identified:

- In the results, 'In contrast, no complex formation was detected in lanes 6-8 which contained at least one mutant protein (Figure 3A).'

→ We modified this sentence like this: In contrast, complex formation was not detected in lanes 6–8, which contained at least one mutant protein (Figure 3A).

-In the results, 'and therefore they were used for tracking KIF1C-mediated subcellular transport in previous18,24,28 and in this study" should be "and therefore they were used for tracking KIF1C-mediated subcellular transport in previous studies18,24,28 and this study"

→ We modified this sentence as suggested.

-In the discussion part, ' Moreover, tail enrichment of Hook3 and KIF1C (Figure 4) and transport of the Hook3-congugated mitochondria cargo to the cell tips (Figure 5) occurred only when the two proteins were co-expressed as wild-type, but not when either one of them was expressed in the binding-disrupted mutant form.'

→ We revised the typo from “congugated” to “conjugated”.

> Reviewer 2

1. It is unclear to me what the "microtubule-anchoring Hook domain" means. The phrase implies that the Hook domain binds directly to microtubules to me, but I am unaware of any evidence of this. Please clarify the meaning of this terminology.

→ We modified this phrase like this: the Hook domain (residues 1–160) that interacts with light intermediate chain 1, a component of the dynein complex (Lee *et al*, 2018).

2. Figure 1 would be much easier to understand if it included a diagram of each full-length protein, highlighting the fragments/domains used in each experiment (similar to the diagram in Fig. 2).

→ In response to the reviewer's comment, we newly added a diagram of each full-length protein as Figure 1A, in which the fragments used in each experiment were highlighted in red.

3. "we found that EGFP-tagged mitochondria were enriched in the tail region (Figure 5B, marked by dashed rectangles". I do not see dashed rectangles in this figure panel. Perhaps the authors meant colored rectangles?

→ We changed the words “dashed rectangles” to “colored rectangles”.

4. The authors use "type I" and "type II" heptad repeat to describe regions of Hook3 but these terms are not standard in the field that I am aware. The authors may want to be clear that they are defining these terms in this work.

→ As the reviewer indicated, we defined the heptad repeat types to describe the Hook3 homodimer structure in the original manuscript. Now we modified the paragraph to show our intention more clearly: “In this study, we defined them as types I (residues 553–601, seven repeats) and II (residues 602–615, two repeats) based on the position of the coiled-coil interface-forming hydrophobic residues: type I at the third and sixth, and type II at the third and seventh (Figure EV2A).”

5. The implication of the competitive interaction between PTPN21 and Hook3 for KIF1C could be more fully elucidated in the discussion. Does the phosphatase activity of PTPN21 play any role?

→ We added a paragraph addressing this topic in the discussion section like this: Previously, it was reported that the interaction between PTPN21 and KIF1C is enhanced by the c-Src kinase-mediated phosphorylation of four tyrosine residues in KIF1C: Tyr 654, Tyr671, Tyr726, and Tyr757 (Saji et al., 2022). Interestingly, c-Src is known as a PTPN21 substrate and is activated through dephosphorylation by PTPN21 (Cardone et al, 2004; Carlucci et al, 2008). Furthermore, among the four KIF1C tyrosine residues, Tyr757 was revealed as the key residue for the Hook3CC3–KIF1CH3BD complex formation via our crystal structure determination, and thus its phosphorylation by c-Src is expected to impair the intermolecular binding (Figure 2C). Taken together, the FERM domain-dependent association of PTPN21 with KIF1C appears to be supported by its amino-terminal phosphatase domain, which can dephosphorylate and activate c-Src. Once activated, c-Src phosphorylates KIF1C, promoting the PTPN21–KIF1C interaction while presumably inhibiting the Hook3–KIF1C binding. Therefore, we propose that the two domains of PTPN21 functionally cooperate to precisely regulate its interaction with KIF1C, which requires further investigation for verification.

6. Fig. 6B is somewhat complicated. There is no step showing the competitive interaction between PTPN21 and Hook3 for KIF1C. Is this assumed to happen between step 1 and step 2? If so, this should probably be labeled. It is unclear to me what "normal" and "depletion" interactions mean in the labeling. I don't fully understand the right side of the cartoon.

➔ First, we supplemented novel descriptions like “Step2. Replacement of PTPN21 with Hook3 in KIF1C binding” and “Dissociated PTPN21” in Figure 6B to show the competitive interaction between PTPN21 and Hook3 for KIF1C more clearly as suggested. Second, we changed the bottom labeling like this: “Cargo transport with Hook3 and KIF1C” and “Cargo transport with Hook3(VE) and KIF1C(YAFA)” to show their meaning more clearly. As our new label describes, the right side of the cartoon shows what happens when the binding-defective Hook3(VE) and KIF1C(YAFA) mutants were expressed in cells (see Figure 5). We also revised Figure 6B overall in response to this reviewer’s comment.

Dear Dr. Ku,

Thank you for submitting your revised manuscript. It has now been seen by one of the original referees.

As you can see, referee finds that the study is significantly improved during revision and recommend publication. However, I need you to address the points below before I can accept the manuscript.

- Please reduce the number of keywords to 5, which is the maximum number we can accommodate.
- Please rename the Declaration of Interests section as 'Disclosure And Competing Interests Statement' and place it after Acknowledgements.
- Please remove the 'Author Contributions' section from the manuscript text.
- We note that there are two dataset citations in the References, whose format should be updated as in the example below:
Hörnberg E, Ylitalo EB, Crnalic S, Antti H, Stattin P, Widmark A, Bergh A, Wikström P (2011) Gene Expression Omnibus GSE29650 (<https://www.ncbi.nlm.nih.gov/geo/query/acc.cgi?acc=GSE29650>). [DATASET]
Hörnberg E, Ylitalo EB, Crnalic S, Antti H, Stattin P, Widmark A, Bergh A, Wikström P (2011) Expression of androgen receptor splice variants in prostate cancer bone metastases is associated with castration-resistance and short survival. PLoS One 6: e19059

In the main text, these datasets should be cited with the prefix "Data ref:" to distinguish them from the reference to the original article that reported the dataset. Example:

"...were grouped based on the relative levels of AR-Vs expressed, mainly AR-V7 (Hörnberg et al, 2011; Data ref: Hörnberg et al, 2011)."

- We note that ORCID iD of the co-corresponding author Dr. Seung Jun Kim is currently missing.
- EMBO Press policy asks for all corresponding authors to link to their ORCID iDs. You can read about the change under "Authorship Guidelines" in the Guide to Authors here:
<https://www.embopress.org/page/journal/14693178/authorguide#authorshipguidelines>

In order to link your ORCID iD to your account in our manuscript tracking system, please do the following:

1. Click the 'Modify Profile' link at the bottom of your homepage in our system.
2. On the next page you will see a box halfway down the page titled ORCID*. Below this box is red text reading 'To Register/Link to ORCID, click here'. Please follow that link: you will be taken to ORCID where you can log in to your account (or create an account if you don't have one)
3. You will then be asked to authorise Wiley to access your ORCID information. Once you have approved the linking, you will be brought back to our manuscript system.

We regret that we cannot do this linking on your behalf for security reasons.

- We find that Appendix Table S1 is better suited to be Table EV1. Please update source file names, titles in the manuscript tracking system, manuscript callouts. Please include its legend in the file.
- We note that 4 movies are uploaded but the nomenclature needs to be Movie EV1-EV4; the legends need to be removed from the manuscript file; each legend should be provided in a readme.txt file and then each movie should be zipped together with its legend, so that we have 4 zip folders Movie EV1-EV4; source file names and manuscript callouts also need to be updated.
- Please remove the Reagents and Tools table from the manuscript text and submit it as a separate word file.
- We note that there is a readme.txt file uploaded as Expanded View listing suppl. material, info regarding the videos - this file is not needed.
- Our production/data editors have asked you to clarify several points in the figure legends - Figure Legends (main + EV):
 - o Please note that the exact p values are not provided in the legend of figure 5D.
 - o Please note that the URL for (Abid Ali et al , 2025 and Siddiqui et al, 2019) data citation is not provided.
- The synopsis image needs to be 550px wide and 300-600px high. When your synopsis image is resized accordingly, the labels are too small to read (please see attached). Please provide a synopsis image with larger labels.

Thank you again for giving us to consider your manuscript for EMBO Reports, I look forward to your minor revision.

Kind regards,

Deniz Senyilmaz Tiebe

--

Deniz Senyilmaz Tiebe, PhD
Senior Scientific Editor

EMBO Reports

Referee #2:

The authors have done a good job responding to my initial comments and I believe they have clarified the manuscript for readers. This work provides interesting insights into cargo-mediated activation of molecular motors and will be of interest to the cell biology community. I support publication of the revised manuscript in EMBO Reports.

Editor's comments

- Please reduce the number of keywords to 5, which is the maximum number we can accommodate.

➔ The number of keywords is now exactly five.

- Please rename the Declaration of Interests section as 'Disclosure And Competing Interests Statement' and place it after Acknowledgements.

➔ The section was renamed as “Disclosure and competing interests statement” and replaced as suggested.

- Please remove the 'Author Contributions' section from the manuscript text.

➔ The 'Author Contributions' section was removed from the manuscript text.

- We note that there are two dataset citations in the References, whose format should be updated as in the example below:

Hörnberg E, Ylitalo EB, Crnalic S, Antti H, Stattin P, Widmark A, Bergh A, Wikström P (2011) Gene Expression Omnibus

GSE29650(<https://www.ncbi.nlm.nih.gov/geo/query/acc.cgi?acc=GSE29650>). [DATASET]

Hörnberg E, Ylitalo EB, Crnalic S, Antti H, Stattin P, Widmark A, Bergh A, Wikström P

(2011) Expression of androgen receptor splice variants in prostate cancer bone metastases is associated with castration-resistance and short survival. PLoS One 6: e19059

In the main text, these datasets should be cited with the prefix "Data ref:" to distinguish them from the reference to the original article that reported the dataset. Example:

"...were grouped based on the relative levels of AR-Vs expressed, mainly AR-V7 (Hörnberg et al, 2011; Data ref: Hörnberg et al, 2011)."

➔ During the second revision, we concluded that only one dataset citation is valid, which is now shown in the reference section like this:

Abid Ali F, Zwetsloot AJ, Stone CE, Morgan TE, Wademan RF, Carter AP, Straube A (2025)

Cryo-EM structure of Fts-Hook3-FHIP1B at 3.2 Å resolution

(<https://www.rcsb.org/structure/8QAT>). [DATASET]

Abid Ali F, Zwetsloot AJ, Stone CE, Morgan TE, Wademan RF, Carter AP, Straube A (2025)

KIF1C activates and extends dynein movement through the FHF cargo adapter. Nat

Struct Mol Biol:Online ahead of print

In the main text, this dataset is cited with the prefix "Data ref:" as suggested like this: (Abid Ali et al, 2025; Data ref: Abid Ali et al, 2025).

• We note that ORCID iD of the co-corresponding author Dr. Seung Jun Kim is currently missing.

➔ The ORCID iD of the co-corresponding author Dr. Seung Jun Kim is now activated as <https://orcid.org/0000-0003-0293-6972>.

• EMBO Press policy asks for all corresponding authors to link to their ORCID iDs. You can read about the change under "Authorship Guidelines" in the Guide to Authors here:

<https://www.embopress.org/page/journal/14693178/authorguide#authorshipguidelines>

In order to link your ORCID iD to your account in our manuscript tracking system, please do the following:

1. Click the 'Modify Profile' link at the bottom of your homepage in our system.
2. On the next page you will see a box halfway down the page titled ORCID*. Below this box is red text reading 'To Register/Link to ORCID, click here'. Please follow that link: you will be taken to ORCID where you can log in to your account (or create an account if you don't have one)
3. You will then be asked to authorise Wiley to access your ORCID information. Once you have approved the linking, you will be brought back to our manuscript system.

We regret that we cannot do this linking on your behalf for security reasons.

➔ All the ORCID iDs of the three corresponding authors are now linked to each of the EMBO reports account.

Title	Dr.	Title	Prof.	Title	Dr.
First Name*	Seung Jun	First Name*	Won Do	First Name*	Bonsu
Middle Name		Middle Name		Middle Name	
Last Name*	Kim	Last Name*	Heo	Last Name*	Ku
ORCID	0000-0003-0293-6972	ORCID	0000-0001-7541-7319	ORCID	0000-0003-1784-8975

- We find that Appendix Table S1 is better suited to be Table EV1. Please update source file names, titles in the manuscript tracking system, manuscript callouts. Please include its legend in the file.

➔ The crystallographic statistics are now represented in “Table EV1” along with the manuscript update.

- We note that 4 movies are uploaded but the nomenclature needs to be Movie EV1-EV4; the legends need to be removed from the manuscript file; each legend should be provided in a readme.txt file and then each movie should be zipped together with its legend, so that we have 4 zip folders Movie EV1-EV4; source file names and manuscript callouts also need to be updated.

➔ The four movies are now called “Movie EV1-EV4”, each of which is zipped together with its legend along with the manuscript update.

- Please remove the Reagents and Tools table from the manuscript text and submit it as a separate word file.

➔ The Reagents and Tools table was removed from the manuscript and submitted as a separate file as suggested.

- We note that there is a readme.txt file uploaded as Expanded View listing suppl. material, info regarding the videos - this file is not needed.

➔ The readme.txt file was removed.

- Our production/data editors have asked you to clarify several points in the figure legends - Figure Legends (main + EV):

- o Please note that the exact *p* values are not provided in the legend of figure 5D.

➔ The missed *p* value, described as $P < 0.0001$ in the manuscript, is now noted precisely ($P = 3.4137 \times 10^{-16}$) in the Figure 5D (right panel).

- o Please note that the URL for (Abid Ali et al , 2025 and Siddiqui et al, 2019) data citation is not provided.

➔ During the second revision, we concluded that only one dataset citation is valid, which is now shown in the reference section like this:

Abid Ali F, Zwetsloot AJ, Stone CE, Morgan TE, Wademan RF, Carter AP, Straube A (2025)

Cryo-EM structure of Fts-Hook3-FHIP1B at 3.2 Å resolution

(<https://www.rcsb.org/structure/8QAT>). [DATASET]

Abid Ali F, Zwetsloot AJ, Stone CE, Morgan TE, Wademan RF, Carter AP, Straube A (2025)

KIF1C activates and extends dynein movement through the FHF cargo adapter. Nat Struct Mol Biol:Online ahead of print

- The synopsis image needs to be 550px wide and 300-600px high. When your synopsis image is resized accordingly, the labels are too small to read (please see attached). Please provide a synopsis image with larger labels.

➔ We modified the synopsis image with larger labels as suggested.

Dr. Bonsu Ku
Korea Research Institute of Bioscience and Biotechnology
Disease Target Structure Research Center
125 Gwahak-ro, Yuseong
Daejeon, Daejeon 34141
Korea, Republic of

Dear Dr. Ku,

Thank you for submitting your revised manuscript. I have now looked at everything and all is fine. Therefore, I am very pleased to accept your manuscript for publication in EMBO Reports.

Congratulations on a nice work!

Kind regards,

Deniz Senyilmaz Tiebe

--

Deniz Senyilmaz Tiebe, PhD
Senior Scientific Editor
EMBO Reports

--
